# BWCache: Accelerating Video Diffusion Transformers through Block-Wise Caching

**Hanshuai Cui[1,2], Zhiqing Tang[2]\*, Zhifei Xu[2], Zhi Yao[1,2], Wenyi Zeng[1], Weijia Jia[2]**

[1]School of Artificial Intelligence, Beijing Normal University, Beijing 100875, China

[2]Institute of Artificial Intelligence and Future Networks, Beijing Normal University, Zhuhai 519087, China

## Abstract

Recent advancements in Diffusion Transformers (DiTs) have established them as the state-of-the-art method for video generation. However, their inherently sequential denoising process results in inevitable latency, limiting real-world applicability. Existing acceleration methods either compromise visual quality due to architectural modifications or fail to reuse intermediate features at proper granularity. Our analysis reveals that DiT blocks are the primary contributors to inference latency. Across diffusion timesteps, the feature variations of DiT blocks exhibit a U-shaped pattern with high similarity during intermediate timesteps, which suggests substantial computational redundancy. In this paper, we propose Block-Wise Caching (BWCache), a training-free method to accelerate DiT-based video generation. BWCache dynamically caches and reuses features from DiT blocks across diffusion timesteps. Furthermore, we introduce a similarity indicator that triggers feature reuse only when the differences between block features at adjacent timesteps fall below a threshold, thereby minimizing redundant computations while maintaining visual fidelity. Extensive experiments on several video diffusion models demonstrate that BWCache achieves up to $2.6\times$ speedup with comparable visual quality. The code is available at https://github.com/hsc113/BWCache.

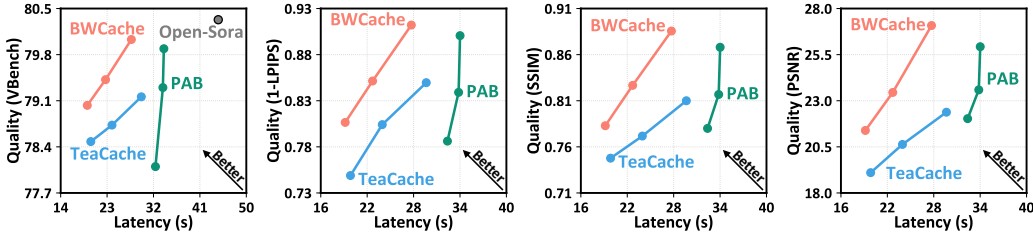

Figure 1: Quality-latency comparisons for video diffusion models. Visual quality versus latency curves are presented for the proposed BWCache method, PAB, and TeaCache using Open-Sora. BWCache demonstrates significantly superior visual quality and efficiency compared to both PAB and TeaCache. Latency is evaluated on a single NVIDIA A800 GPU for generating 51 frames, 480P videos.

## 1 Introduction

Diffusion Models (DMs) Song & Ermon (2019); Ho et al. (2020); Dhariwal & Nichol (2021) have emerged as the predominant technology in Generative AI (GenAI) due to their ability to generate high-fidelity images and videos Rombach et al. (2022). Early DMs primarily adopted the U-Net architecture Ronneberger et al. (2015); Ramesh et al. (2022); Saharia et al. (2022); Blattmann et al. (2023). Recently, the success of Sora Brooks et al. (2024) has spurred the adoption of the Diffusion Transformer (DiT) architecture Peebles & Xie (2023); Bao et al. (2023) for video generation,

---

\*Corresponding author

Figure 2: Overview of the BWCache. An indicator based on adjacent timestep differences in block features determines whether to reuse the cache. If conditions are met, subsequent timesteps reuse cached blocks; otherwise, blocks are recomputed and the cache updated.

achieving state-of-the-art performance. Video generation with DiT requires the sequential computation of multiple DiT blocks, including spatial and temporal DiT blocks Ho et al. (2022) However, these blocks must be executed in sequence, which considerably slows down inference. As a result, accelerating DiT inference has become one of the most pressing challenges faced by GenAI.

Traditional acceleration techniques for DMs primarily focus on architectural modifications such as distillation, pruning, and quantization Wang et al. (2024); Chu et al. (2024); Yang et al. (2024); Zhang et al. (2024a); Kang et al. (2024); Zhou et al. (2024). Although these methods reduce model complexity, they also compromise generation quality. More importantly, they require substantial computational resources and extensive training data to fine-tune, which is impractical for large pre-trained DMs. A training-free alternative is to cache the intermediate features of DMs to accelerate inference Ma et al. (2023); Zhang et al. (2024b); Zhao et al. (2024b); Kahatapitiya et al. (2024); Liu et al. (2024). By reusing these intermediate features across timesteps, these approaches eliminate redundant computations and significantly improve generation efficiency.

Recent work has explored block-level temporal stability in DiTs. Skip-DiT Chen et al. (2025) addresses feature instability by introducing Long-Skip-Connections to stabilize DiT block features, enabling efficient caching. However, this method requires modifying the model architecture and retraining from scratch, which is impractical for large pre-trained models and limits their applicability. ProfilingDiT Ma et al. (2025b) indicates that redundancy is not uniformly distributed across timesteps. While their work does not focus on block-wise analysis in DiTs, their findings are conceptually aligned with our observation that mid-timestep regions exhibit higher redundancy. However, this method requires storing extensive intermediate features across multiple timesteps for profiling, leading to significant memory overhead.

Despite the proven effectiveness of caching in accelerating DM inference, two critical challenges remain unresolved. First, DiT is a complex structure, considering caching at a coarse granularity (e.g., timesteps) Lv et al. (2025); Liu et al. (2024); Ma et al. (2025b) may result in the loss of essential information, while considering caching at a fine granularity (e.g., attention) Kahatapitiya et al. (2024); Zhao et al. (2024b); Yuan et al. (2024); Zhang et al. (2025) often fails to deliver significant acceleration. Thus, the first challenge is to identify which features are appropriate for caching. Second, many existing methods assume high similarity between features across adjacent timesteps, sharing features between them to speed up inference Zhao et al. (2024b); Lv et al. (2025); Wimbauer et al. (2024); Selvaraju et al. (2024); Liu et al. (2025c); Chen et al. (2025). In reality, however, feature similarity varies greatly depending on the generation task and specific inference process, and naive feature reuse can often degrade output details. Therefore, the second challenge is to determine when cached features should be reused or updated.

In this paper, we propose Block-Wise Caching (BWCache), a training-free method to accelerate DiT-based video generation. This method can be seamlessly integrated into most DiT-based models as a plug-and-play component during the inference phase. The core idea is to cache the features from all DiT blocks at certain diffusion timesteps and reuse them across several subsequent steps. Specifically, as shown in Figure 2, DiT-based video generation typically involves providing a text prompt and initializing latent noise, which is then refined through multiple iterative denoising steps, and finally reconstructed into video frames using a VAE decoder. To determine when to reuse cached features, we introduce a similarity indicator based on the differences between block features from adjacent timesteps. Periodic recomputation is applied to mitigate potential latent drift. Our method is training-free and memory-efficient, using a lightweight similarity indicator to dynamically determine cache reuse without extensive feature storage, making it directly applicable to existing pre-trained models without architectural modifications. We evaluate BWCache across several video

diffusion models, including Open-Sora Zheng et al. (2024), Open-Sora-Plan Lin et al. (2024a), Latte Ma et al. (2025a), Wan 2.1 Wan et al. (2025), and HunyuanVideo Kong et al. (2024). As shown in Figure 1, BWCache achieves superior performance at comparable computational costs to TeaCache Liu et al. (2024) and PAB Zhao et al. (2024b). The main contributions are summarized as follows.

- We analyze the components of DiT-based models during the denoising process, revealing the dynamics within DiT blocks under different generation tasks and highlighting their similarities across specific diffusion timesteps.
- We propose BWCache, a training-free method compatible with DiT-based models that caches DiT block features and reuses them across multiple diffusion timesteps, thereby accelerating video generation inference.
- Experiments demonstrate that our method outperforms existing baselines in terms of inference speed and video generation quality, justified through both ablation studies and qualitative comparisons.

## 2 RELATED WORK

### 2.1 DIFFUSION MODEL

Early video generation models, such as Variational Autoencoders (VAEs) Kingma et al. (2013) and Generative Adversarial Networks (GANs) Goodfellow et al. (2014), face several challenges, including blurry outputs, training instability, and limited text-video alignment. The emergence of DMs Ho et al. (2020); Dhariwal & Nichol (2021) addresses these limitations by achieving high-quality and diverse video generation through a gradual denoising process. Initially, these models employed U-Net architectures, which demonstrated impressive results in both image and video generation tasks Ramesh et al. (2022); Chen et al. (2024a); Wei et al. (2024). More recently, advancements in DiT-based architectures Peebles & Xie (2023); Bao et al. (2023) have further improved scalability and generalization over traditional U-Net-based DMs. The success of the Sora model Brooks et al. (2024), which utilizes the DiT architecture for long-form video generation, has led to a growing body of research adopting DiT as the noise estimation network Hatamizadeh et al. (2024); Wang et al. (2025); Li et al. (2025).

### 2.2 DIFFUSION MODEL ACCELERATION

Despite the remarkable performance of DMs, their high inference costs limit their applicability in real-world applications. Existing acceleration techniques for DMs can generally be categorized into two groups. The first category relies on architectural changes, such as distillation simplifying networks Kang et al. (2024); Zhou et al. (2024); Salimans et al. (2024), pruning removes redundant parameters Yang et al. (2024); Zhang et al. (2024a); Castells et al. (2024), quantization reduces precision Wang et al. (2024); Chu et al. (2024); Huang et al. (2024a), and others. However, these approaches often require additional resources for fine-tuning or optimization. The second group consists of training-free methods that accelerate inference by caching intermediate features of diffusion models. DeepCache Ma et al. (2023) accelerates diffusion models by caching high-level features from sequential denoising timesteps. T-GATE Zhang et al. (2024b) enhances image generation efficiency by caching attention outputs. PAB Zhao et al. (2024b) leverages the attention redundancy to significantly accelerate the diffusion process. AdaCache Kahatapitiya et al. (2024) accelerates video generation by selectively caching computations based on video content. TeaCache Liu et al. (2024) utilizes timestep embeddings to dynamically determine which computations to cache to accelerate inference. Although these training-free methods enhance diffusion efficiency, they still face significant challenges in balancing generation quality with computational cost.

## 3 METHODOLOGY

### 3.1 PRELIMINARIES

**Denoising Diffusion Models.** Consider the data $\mathbf{x}_0 \sim q(\mathbf{x})$ sampled from a real distribution. The forward diffusion process incrementally adds Gaussian noise over $T$ steps, producing a sequence $\mathbf{x}_1, \ldots, \mathbf{x}_T$. As $t$ increases, $\mathbf{x}_t$ progressively loses distinguishable features, asymptotically

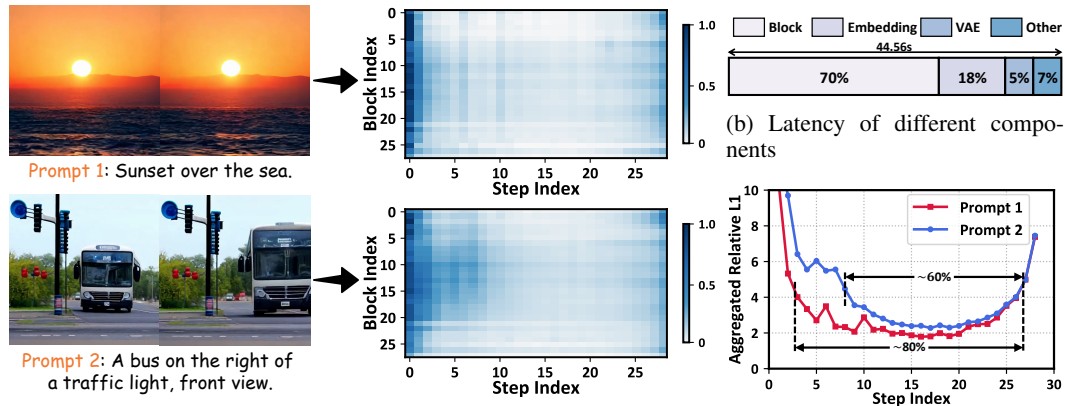

(b) Latency of different components

(a) Two prompts and corresponding generated videos. The heatmaps track the denoising evolution across 28 blocks over 30 steps.

(c) Quantitative block difference

Figure 4: Analysis of the block in the DiT-based model.

approaching an isotropic Gaussian distribution when $T \rightarrow \infty$. With the Markov chain assumption, it is expressed as:

$$q(\mathbf{x}_t \mid \mathbf{x}_{t-1}) := \mathcal{N}(\mathbf{x}_t; \sqrt{1 - \beta_t}\, \mathbf{x}_{t-1},\, \beta_t I), \tag{1}$$

where $q(\mathbf{x}_t|\mathbf{x}_{t-1})$ is the posterior probability. The step sizes are controlled by a noise schedule $\beta_1, \ldots, \beta_T$. The reverse process samples from $q(\mathbf{x}_{t-1}|\mathbf{x}_t)$ to reconstruct data from noise $\mathbf{x}_T \sim \mathcal{N}(\mathbf{0}, \mathbf{I})$. Similarly, the backward diffusion process can be written as:

$$p_\theta(\mathbf{x}_{t-1} \mid \mathbf{x}_t) := \mathcal{N}(\mathbf{x}_{t-1}; \mu_\theta(\mathbf{x}_t, t),\, \Sigma_\theta(\mathbf{x}_t, t)), \tag{2}$$

where $\mu_\theta$ and $\Sigma_\theta$ are learned parameters defining the mean and covariance, and $p_\theta(.)$ denotes the probability of observing $\mathbf{x}_{t-1}$ given $\mathbf{x}_t$.

**DiT Block.** The DiT block Ho et al. (2022); Peebles & Xie (2023) follows the two-stage Transformer design to modulate the dynamics of diffusion features across both space and time. Each denoising step involves sequential processing through $N$ consecutive DiT blocks, which is illustrated in Figure 3. For the $i$-th DiT block ($i \in [1, N]$) in timestep $t$, given an input hidden state $h_{t,i}$, the block first performs Adaptive Layer Normalization (AdaLN) Xu et al. (2019) followed by multi-head self-attention and cross-attention:

$$h'_{t,i} = \text{Attention}(\text{AdaLN}(h_{t,i})) + h_{t,i}, \tag{3}$$

where the residual connection preserves input information. A subsequent AdaLN-modulated MLP enhances feature expression:

$$h''_{t,i} = \text{MLP}(\text{AdaLN}(h'_{t,i})) + h'_{t,i}. \tag{4}$$

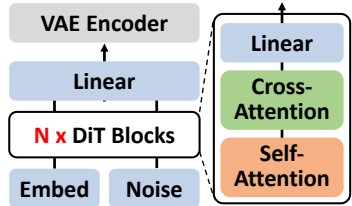

Figure 3: Overview of DiT-based video generation models.

The output hidden state $h''_{t,i}$ then becomes the input to the next DiT block (e.g., $h_{t,i+1} = h''_{t,i}$).

## 3.2 ANALYSIS

**Costs of Diffusion Models.** The video generation process using DMs comprises multiple computational components, including the DiT blocks, timestep and text embeddings, VAE, and others. Figure 4(b) shows the time distribution for each part of the video generation process. As illustrated, the DiT blocks require considerably more computation time than the other components. Furthermore, this proportion increases further as the video length and resolution grow, posing a substantial challenge to the efficiency of video generation.

**Block Feature Variation.** To better understand the internal behavior of DiT blocks, we introduce the relative L1 distance Liu et al. (2024) to measure the feature variations of each block between

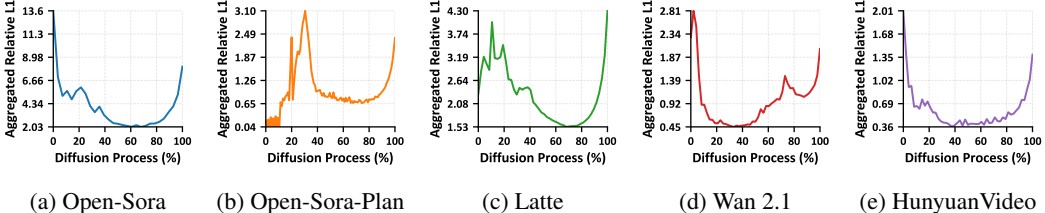

|     |     |     |     |     |
| --- | --- | --- | --- | --- |
| (a) Open-Sora | (b) Open-Sora-Plan | (c) Latte | (d) Wan 2.1 | (e) HunyuanVideo |

Figure 5: Aggregation relative L1 of different models.

adjacent timesteps. For the $i$-th block at the $t$-th step, the block-wise relative L1 distance is calculated as follows:

$$\text{L1}_{\text{rel}}(h_{t,i}) = \frac{\|h_{t,i} - h_{t+1,i}\|_1}{\|h_{t+1,i}\|_1}. \tag{5}$$

A smaller $\text{L1}_{\text{rel}}$ value indicates minimal feature variation within the block between adjacent timesteps, while a larger value shows significant changes. In Figure 4(a), we present two videos generated from different prompts. The video generated with Prompt 1 depicts a relatively static scene, whereas the video generated with Prompt 2 exhibits more dynamic content (e.g., a bus moving closer from a distance). We employ heatmaps to quantify the changes in block features during the generation process for both videos. In these heatmaps, the color intensity reflects the relative degree of block feature variation at different timesteps, with darker regions indicating greater differences. From the figures, we observe two key insights: 1) the colors are darker in the initial and final timesteps, indicating that block features change across timesteps; 2) compared to the figure below, the darker regions in the figure above occupy a smaller proportion, suggesting that different prompts exert varying influences on the features of the blocks.

**Similarity and Diversity.** To gain a more intuitive understanding of the changes in block feature at different timesteps, we aggregate the block-level feature differences within each timestep and define the aggregated relative L1 distance. The equation is as follows:

$$\text{ARL1}(t) = \sum_{n=1}^{N} \text{L1}_{\text{rel}}(h_{t,i}), \tag{6}$$

where $N$ is the total number of DiT blocks. $\text{ARL1}(t)$ quantifies the feature change at a given diffusion step $t$ by summing the relative L1 distances across all blocks.

Figure 4(c) illustrates the aggregated relative L1 distance for two prompts. Generally, the differences in block features between adjacent diffusion timesteps exhibit a U-shaped pattern. In the early timesteps, the L1 distance is high, indicating rapid changes in noise predictions. The middle timesteps gradually stabilize, showing considerable redundancy in block calculations. While in the final timesteps, the aggregated relative L1 distance increases, suggesting a transition from structured noise to high-fidelity video. Notably, for simpler or more static scenes, the feature variations among blocks stay minimal, explaining why Prompt 1 exhibits a longer duration of stable timesteps compared to Prompt 2. More detailed analysis of block feature behavior in Appendix A.

To further validate the consistency of this U-shaped pattern across different architectures and training methodologies, we analyze the aggregated relative L1 distance across five video diffusion models, as shown in Figure 5. The theoretical foundation for this U-shaped pattern can be understood through the lens of frequency domain analysis Qian et al. (2024). During early timesteps, low-frequency components recover, causing high feature variation (left U-shape). Middle timesteps stabilize these, minimizing changes (U-bottom). Final timesteps recover high-frequency details, increasing variation again (right U-shape). However, we observe that some models exhibit more pronounced early-stage feature oscillations compared to other models. This can be attributed to noise schedule and sampling implementation issues Lin et al. (2024b). Specifically, Open-Sora-Plan uses non-linear samplers (i.e. PNDM) that do not start from the last timestep ($t = T$), creating a training-inference discrepancy where the model is forced to handle inputs that deviate from pure Gaussian noise, potentially causing initial timestep feature prediction instability and oscillations.

### 3.3 BWCACHE

As illustrated in Figure 4, intermediate stages of video generation frequently exhibit redundant block computations. To minimize computational redundancy and accelerate inference, we propose Block-Wise Caching (BWCache). Instead of computing each DiT block at every timestep, we selectively reuse block features cached from the previous timestep. We employ the average of the accumulated relative L1 distance as a similarity indicator to determine when to reuse the cached blocks:

$$\sum_{n=1}^{N} \text{L1}_{\text{rel}}(h_{t,i})/N < \delta, \tag{7}$$

where $\text{L1}_{\text{rel}}$ is defined in Eq.(5), and $\delta$ is the similarity threshold. The threshold $\delta$ represents an acceptable level of variation, and when features change more, fewer blocks are cached, naturally adapting to the scene dynamics. Specifically, at each timestep, we calculate the relative L1 distance between each DiT block and its counterpart from the preceding timestep. After each timestep, we cache all block features and calculate the arithmetic mean of their relative L1 distances. If this mean is less than the indicator threshold $\delta$, block computations can be skipped for subsequent timesteps starting from $t - 1$, allowing the reuse of cached features; otherwise, the block will be recomputed. A smaller threshold $\delta$ reduces cache reuse, while a larger threshold speeds up video generation, but may adversely affect video quality. The appropriate threshold $\delta$ should be chosen to optimize inference speed without compromising visual quality.

Reusing DiT blocks in the final diffusion timesteps significantly degrades visual fidelity, as these timesteps represent the critical phase where the latent space transitions from structured noise to a high-quality video. Once BWCache is triggered at the $k$-th step, caching reuse is restricted to the first half of the remaining steps, while the second half is explicitly computed. Therefore, the last $k/2$ steps are always recomputed, ensuring thorough refinement during the most sensitive stage of generation.

### 3.4 PERIODIC CACHE RECOMPUTATION

Continuously reusing the same block may result in latent drift and gradually erase fine-grained details Liu et al. (2025a). To prevent this cache stagnation, we adopt the progressive computation strategy from PAB Zhao et al. (2024b): within the caching interval, each DiT block is periodically recomputed at a defined reuse interval $R$. Specifically, after computing the block at time step $t$, its features are cached and reused for the subsequent $R$ steps (i.e., $[t - 1, t - R]$), and the block is then updated at step $t - R - 1$. This process is formalized as:

$$\mathcal{O}_H = \{\ldots, \underbrace{h_t''}_{\text{cached}}, \underbrace{h_t'', \ldots, h_t''}_{\text{reuse } R \text{ steps}}, \underbrace{h_{t-R-1}''}_{\text{cached}}, \ldots\}, \tag{8}$$

where $\mathcal{O}_H$ denotes the output of the DiT block across all steps, and $h_t''$ represents the DiT block output calculated at step $t$, which can be determined using Eq.(4).

## 4 EXPERIMENTAL RESULTS

### 4.1 EXPERIMENT SETUP

**Base Models and Compared Methods.** We apply the proposed acceleration techniques to various video generation diffusion models, including Open-Sora Zheng et al. (2024), Open-Sora-Plan Lin et al. (2024a), Latte Ma et al. (2025a), Wan 2.1 Wan et al. (2025), and HunyuanVideo Kong et al. (2024). We compare our method with recent effective video generation techniques such as $\Delta$-DiT Chen et al. (2024b), T-GATE Zhang et al. (2024b), PAB Zhao et al. (2024b), and TeaCache Liu et al. (2024) to highlight our advantages. The inference configs of base models are shown in Appendix B.

**Evaluation Metrics and Datasets.** To evaluate the performance of video generation acceleration methods, we primarily focus on two aspects: visual quality and inference efficiency. We utilize VBench Huang et al. (2024b), LPIPS Zhang et al. (2018), PSNR, and SSIM Wang & Bovik (2002) for visual quality assessment. VBench is a comprehensive benchmarking suite for video generation

| Model | Method | Visual Quality | | | | Efficiency | |
|-------|--------|--------|--------|--------|--------|--------|--------|
| | | VBench↑ | LPIPS↓ | SSIM↑ | PSNR↑ | Speedup↑ | Latency(s)↓ |
| **Open-Sora** | Original | 80.33% | - | - | - | 1× | 44.56 |
| | Δ-DiT | 78.21% | 0.5692 | 0.4811 | 11.91 | 1.03× | 43.26 |
| | T-GATE | 77.61% | 0.3495 | 0.6760 | 15.50 | 1.19× | 37.45 |
| | PAB | 78.10% | 0.2134 | 0.7798 | 22.02 | 1.38× | 32.38 |
| | TeaCache | 79.16% | 0.1496 | 0.8104 | 22.39 | 1.50× | 29.64 |
| | FasterCache | 79.21% | 0.1165 | 0.8435 | 23.99 | 1.35× | 32.03 |
| | BWCache | **80.03%** | **0.0879** | **0.8854** | **27.05** | **1.61×** | **27.68** |
| **Open-Sora-Plan** | Original | 80.88% | - | - | - | 1× | 99.65 |
| | Δ-DiT | 77.55% | 0.5388 | 0.3736 | 13.85 | 1.01× | 98.66 |
| | T-GATE | 80.15% | 0.3066 | 0.6219 | 18.80 | 1.18× | 84.45 |
| | PAB | 80.30% | 0.2781 | 0.6627 | 19.49 | 1.36× | 73.41 |
| | TeaCache | 80.32% | 0.1965 | 0.7502 | 21.50 | **4.41×** | **22.62** |
| | FasterCache | 80.61% | 0.1190 | 0.8103 | 24.06 | 1.51× | 65.79 |
| | BWCache | **80.82%** | **0.1001** | **0.8435** | **25.87** | 2.24× | 44.49 |
| **Latte** | Original | 78.95% | - | - | - | 1× | 26.90 |
| | Δ-DiT | 52.00% | 0.8513 | 0.1078 | 8.65 | 1.02× | 26.37 |
| | T-GATE | 75.42% | 0.2612 | 0.6927 | 19.55 | 1.13× | 23.81 |
| | PAB | 76.32% | 0.4625 | 0.5964 | 17.03 | 1.21× | 22.16 |
| | TeaCache | 77.40% | 0.1969 | 0.7606 | 22.19 | 1.86× | 14.46 |
| | FasterCache | 78.21% | 0.2442 | 0.7304 | 20.66 | 1.38× | 19.56 |
| | Skip-DiT | 75.76% | 0.1403 | 0.8087 | 25.50 | 1.65× | 16.28 |
| | BWCache | **78.28%** | **0.1399** | **0.8181** | **26.46** | **1.90×** | **14.16** |
| **Wan 2.1** | Original | 82.17% | - | - | - | 1× | 912 |
| | PAB | 80.04% | 0.1623 | 0.7643 | 21.32 | 1.19× | 767 |
| | TeaCache | 81.73% | 0.2407 | 0.6593 | 18.62 | 1.41× | 644 |
| | BWCache | **81.99%** | **0.0782** | **0.8539** | **25.86** | **2.00×** | **457** |
| **HunyuanVideo** | Original | 82.29% | - | - | - | 1× | 1122 |
| | PAB | 81.73% | 0.1045 | 0.8341 | 26.69 | 1.08× | 1039 |
| | TeaCache | 82.13% | 0.1630 | 0.8052 | 24.37 | 2.27× | 493 |
| | BWCache | **82.48%** | **0.0794** | **0.8903** | **29.91** | **2.60×** | **433** |

Table 1: Comparison of visual quality and efficiency on a single GPU. Video generation specifications: Open-Sora (51 frames, 480P), Open-Sora-Plan (65 frames, 512×512), Latte (16 frames, 512×512), Wan 2.1 (81 frames, 480P), HunyuanVideo (129 frames, 544P). LPIPS, SSIM, and PSNR are calculated against the original model results.

models. LPIPS, PSNR, and SSIM measure the similarity between videos generated by the accelerated methods and the original models. Several evaluation metrics are detailed in Appendix C. To assess inference efficiency, we use the inference latency as metrics. In our main experiments, we generate over 1,000 videos for each model under each baseline using prompts sourced from Open-Sora gallery Lin et al. (2024a), VBench benchmark Huang et al. (2024b), and T2V-CompBench Sun et al. (2024).

**Implementation Details.** All experiments are conducted on NVIDIA A800 80GB GPUs using PyTorch. FlashAttention Dao et al. (2022) is enabled by default across all experiments. Unless otherwise specified, ablation studies and qualitative analyses generate 51 frames, 480P videos in 30 steps. $\delta$ is set to 0.15 and reuse interval $R$ is set to 10% of total timesteps by default.

## 4.2 MAIN RESULTS

**Quantitative Comparison.** The experimental results in Table 1 demonstrate the effectiveness of BWCache across multiple DiT-based video generation models. Our proposed BWCache outperforms existing optimization methods in both visual quality and efficiency. For instance, on Open-Sora, BWCache achieves advanced visual metrics while reducing latency by 1.61× compared to the original model. Notably, BWCache surpasses the training-free TeaCache method and maintains competitive acceleration performance. The block-wise method of BWCache leverages DiT block redundancies more effectively without sacrificing visual details. The timestep-level caching of TeaCache is less adaptive to intra-timestep variations, leading to higher quality degradation. It is worth noting that TeaCache achieves faster acceleration than BWCache on the Open-Sora-Plan model. This is because there is significant variation in block features during the early inference stages of the Open-Sora-Plan model. To maintain high-fidelity output quality, BWCache strategically skips

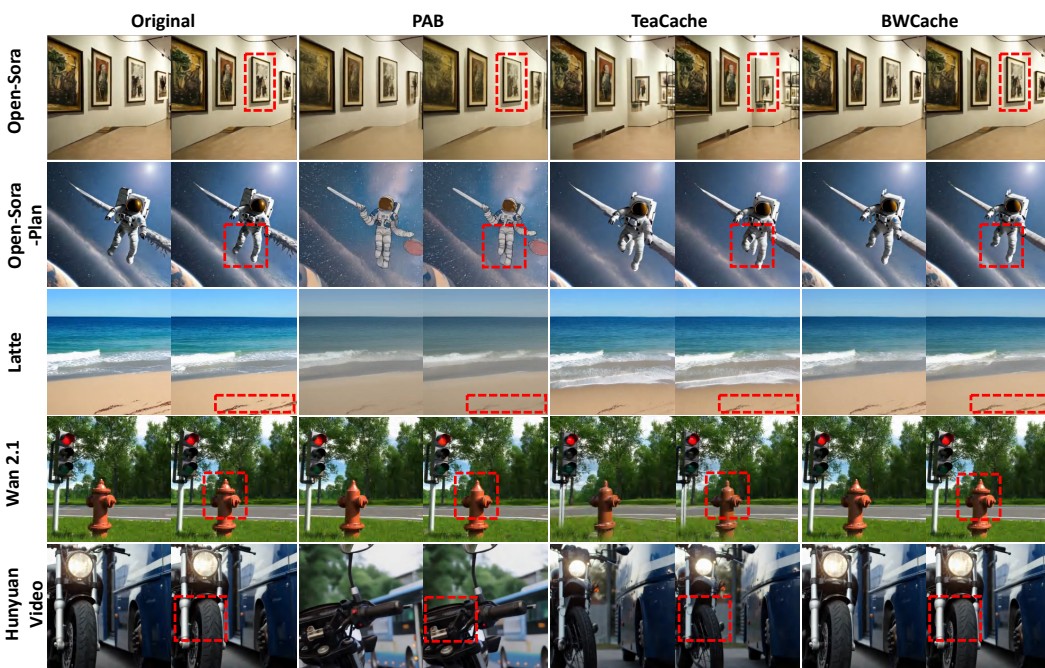

Figure 6: Comparison of visual quality with the compared method. BWCache outperforms PAB and TeaCache in both visual quality and efficiency.

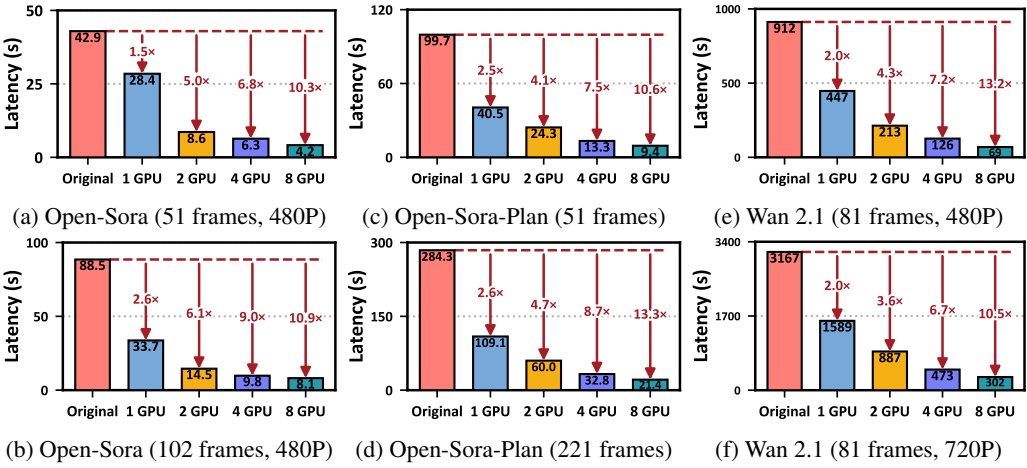

Figure 7: Inference efficiency of BWCache at different video lengths and resolutions.

caching during the first few denoising steps where feature oscillations are most pronounced (as visualized in Figure 5b), thereby sacrificing some acceleration potential in favor of generation quality. We also compare with Skip-DiT Chen et al. (2025) on the Latte model, as shown in Table 1. While Skip-DiT achieves competitive acceleration ($1.65\times$ speedup), it requires architectural modifications and model retraining, which significantly limits its practical applicability. The Skip-DiT repository only provides a pre-trained model for Latte, and according to their documentation, training requires approximately 8 H100 GPUs for about one week with 300k text-video pairs. This substantial training requirement makes it impractical to evaluate Skip-DiT on other models or compare it fairly across different architectures. We further compare with FasterCache, which is another training-free caching-based acceleration method. The official FasterCache repository provides support for Open-Sora, Open-Sora-Plan, and Latte models. BWCache consistently outperforms FasterCache across all three models in terms of visual quality metrics. More experiments can be found in Appendix D.

| Method | 1 × A800 | 2 × A800 | 4 × A800 | 8 × A800 |
|---|---|---|---|---|
| **Open-Sora (204 frames, 480P)** | | | | |
| original | 190.2(1×) | 71.3(2.7×) | 37.7(5.1×) | 24.1(7.9×) |
| PAB | 150.7(1.2×) | 55.3(3.3×) | 29.8(6.1×) | 19.8(9.1×) |
| TeaCache | 114.0(1.7×) | 47.0(4.0×) | 24.6(7.7×) | 14.4(13.2×) |
| BWCache | **78.1(2.4×)** | **30.3(6.3×)** | **16.9(11.3×)** | **11.1(17.2×)** |
| **Open-Sora-Plan (221 frames, 512×512)** | | | | |
| original | 284.3(1×) | 154.3(1.8×) | 81.2(3.5×) | 47.1(6.0×) |
| PAB | 211.4(1.3×) | 112.5(2.5×) | 60.0(4.7×) | 34.9(8.1×) |
| TeaCache | **48.2(5.9×)** | **26.9(10.6×)** | **15.9(17.9×)** | **10.1(28.1×)** |
| BWCache | 109.1(2.6×) | 60.0(4.7×) | 32.8(8.7×) | 21.4(13.3×) |
| **Latte (16 frames, 512×512)** | | | | |
| original | 25.9(1×) | 15.1(1.7×) | 9.3(2.8×) | 8.9(2.9×) |
| PAB | 21.0(1.2×) | 12.2(2.1×) | 8.3(3.1×) | 7.9(4.0×) |
| TeaCache | 17.5(1.5×) | 7.3(3.5×) | 5.6(4.66×) | 5.0(5.2×) |
| BWCache | **14.4(1.8×)** | **7.3(3.6×)** | **4.5(5.8×)** | **4.0(6.5×)** |

Table 2: Inference efficiency when scaling to multiple GPUs with DSP.

| Method | Latency(s)↓ | LPIPS↓ | SSIM↑ | PSNR↑ |
|---|---|---|---|---|
| original, 19 steps | 29.20 | 0.3139 | 0.6922 | 16.39 |
| BWCache, 30 steps | 27.68 | 0.0879 | 0.8856 | 27.08 |

Table 3: Caching mechanism v.s. inference steps.

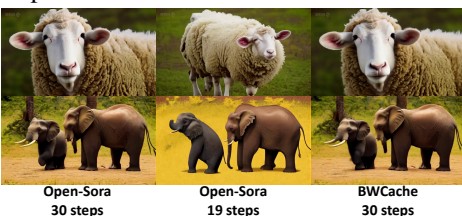

Open-Sora 30 steps    Open-Sora 19 steps    BWCache 30 steps

Figure 8: Accelerate video generation by reducing inference steps and using caching respectively.

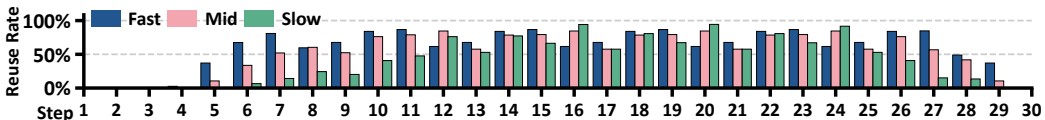

Figure 9: Visualization for the block reuse rate at each step.

**Visual Quality Comparison.** As shown in Figure 6, we visualize the videos generated by our method compared to the original model, PAB, and TeaCache. The results demonstrate that BW-Cache outperforms other methods in visual quality with lower latency. More visual results are illustrated in Appendix F.

### 4.3 SCALING ABILITIES

**Scaling to Multiple GPUs.** Table 2 summarizes the inference efficiency of various methods when scaled across multiple GPUs using Dynamic Sequence Parallelism (DSP) Zhao et al. (2024a). Utilizing the DSP method, GPUs are interconnected via high-bandwidth NVLink, which provides low-latency, high-throughput communication for distributed workloads. A single GPU often cannot utilize a full batch size due to memory constraints, leading to suboptimal hardware utilization. In contrast, the DSP method overcomes this by employing dimension-aware sharding and efficient all-to-all operations to minimize communication overhead, thereby enabling more effective scaling. The experiments, conducted using several models, reveal that BWCache consistently outperforms both the original model, PAB, and TeaCache in terms of latency reduction across all tested scenarios. Specifically, BWCache achieves the lowest latencies in almost all cases, demonstrating significant speed-ups as the number of GPUs increases. These results validate BWCache as a universally effective acceleration method for DiT-based models under multi-GPU deployments.

**Performance at Different Length and Resolution.** Figure 7 demonstrates the inference efficiency of BWCache across various video lengths and resolutions using Open-Sora, Open-Sora-Plan, and Wan 2.1 models. The results show that latency increases significantly as the length and resolution of the video increase, but BWCache consistently performs better than the original model in all configurations. BWCache exhibits particularly notable acceleration advantages in processing high-resolution long videos. For example, with 8 GPUs, BWCache attains a latency of just 11.08 seconds for the Open-Sora model with 204 frames at 480P, representing a remarkable 17.16× speed-up compared to the base model. This trend is also reflected in other configurations, where BWCache consistently provides substantial improvements in efficiency. Overall, these results validate its capability to improve inference speeds across various video lengths and resolutions, making it an efficient solution for real-time video generation.

| Threshold | Reuse Rate | VBench↑ | LPIPS↓ | SSIM↑ | PSNR↑ |
|---|---|---|---|---|---|
| 0.25(Fast) | 59.00% | 79.03% | 0.1935 | 0.7829 | 21.39 |
| 0.20(Mid) | 53.05% | 79.42% | 0.1486 | 0.8267 | 23.45 |
| 0.15(Slow) | 41.38% | 80.03% | 0.0879 | 0.8854 | 27.05 |

Table 4: Impact of different reuse rates.

| Interval | Latency(s)↓ | VBench↑ | LPIPS↓ | SSIM↑ | PSNR↑ |
|---|---|---|---|---|---|
| 5% | 34.70 | 80.28% | 0.0451 | 0.9250 | 30.72 |
| 10% | 27.68 | 80.03% | 0.0879 | 0.8854 | 27.05 |
| 15% | 27.33 | 79.84% | 0.1065 | 0.8709 | 26.13 |
| 20% | 25.97 | 79.48% | 0.1415 | 0.8465 | 24.82 |
| 25% | 25.76 | 79.31% | 0.1567 | 0.8360 | 24.32 |

Table 5: Impact of different reuse intervals.

## 4.4 ABLATION STUDIES

**Reducing Inference Steps.** When generating videos using the DiT model, the impact of the inference steps must be considered. In this paper, the default setting for the inference steps in Open-Sora is 30. To accelerate video generation, the number of inference steps can be reduced. We compare the performance of the original model with reduced inference steps and BWCache in terms of visual quality and latency. The results show that, while producing similar latencies, BWCache maintains excellent visual quality, as shown in Figure 8. Specific data on visual quality and efficiency is detailed in Table 3. Overall, when the number of inference steps is reduced, sampling trajectories deviate from the theoretical optimal path, which amplifies accumulated errors in the noise estimation network. This results in issues such as blurred video details and reduced color saturation.

**Quality-Efficiency Trade-Off.** Figure 1 compares the quality-latency trade-off of BWCache, PAB, and TeaCache. The indicator threshold $\delta$ in Eq.(7) is 0.15, 0.20, and 0.25. The results clearly demonstrate that BWCache outperforms PAB and TeaCache in visual quality across all evaluated metrics, indicating a notable improvement in the fidelity of video generation. For each metric, BWCache not only achieves higher quality scores but also maintains lower latency, which reflects its efficient processing capabilities. Overall, BWCache achieves improved visual quality at lower latency than PAB and TeaCache.

**Reuse Rate.** Table 4 illustrates how varying the reuse rates affects performance. It reveals that the "Fast" configuration achieves the highest reuse rate at 59.00%, resulting in the lowest latency of 19.16 seconds. The "Slow" configuration achieves a reuse rate of 41.38%, but the visual quality is the highest among all configurations. Visualized results in Figure 9 complement these findings by providing the reuse rates of blocks across each timestep for various thresholds. This figure illustrates the dynamic nature of block reuse at different timesteps, with evident fluctuations that suggest varying block reuse efficiency at different stages of video generation. In general, the block features in the middle stage of the generation change less, and a high reuse rate is obtained, which is consistent with our previous analysis.

**Reuse Interval.** Table 5 outlines various reuse intervals, ranging from 5% to 25% of all timesteps, and details their corresponding latency and quality metrics. As the reuse interval increases, the latency decreases, indicating enhanced generating efficiency. Conversely, while the latency improves with increased reuse interval, there is a slight degradation in quality metrics. For instance, VBench scores range from 80.28% at 5% to 79.31% at 25%, indicating a gradual decline in fidelity. This demonstrates a trade-off between latency and visual quality, where longer reuse intervals can effectively accelerate generation but may destroy the details in the output.

## 5 CONCLUSION

In this paper, we introduced BWCache, a novel caching method designed to accelerate video Diffusion Transformers. By analyzing feature dynamics in DiT blocks, we established that block features across timesteps had significant computational redundancy. BWCache addressed this by employing dynamic block-wise caching and a similarity indicator, which selectively reused block features when their differences fell below a predefined threshold. Periodic recomputation was employed to mitigate potential latent drift. Extensive experiments demonstrated that BWCache achieved robust efficiency and visual quality across diverse generation models, sampling schedules, and video parameters. These results highlighted its potential for real-world applications. The current indicator threshold was preset, and future work will dynamically adjust the threshold according to different generation tasks.

ACKNOWLEDGEMENTS

This work was supported in part by the National Natural Science Foundation of China (NSFC) under Grant 62272050 and Grant 62302048; in part by the Guangdong Key Lab of AI and Multi-modal Data Processing, Beijing Normal-Hong Kong Baptist University (BNBU), Zhuhai under 2023-2024 grants sponsored by Guangdong Provincial Department of Education; in part by the Institute of Artificial Intelligence and Future Networks (BNU-Zhuhai) and Engineering Center of AI and Future Education, Guangdong Provincial Department of Science and Technology, China; in part by Zhuhai Science-Tech Innovation Bureau under Grant No. 2320004002772; and in part by the Interdisciplinary Intelligent Supercomputing Center of Beijing Normal University (Zhuhai).

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

## A   BLOCK-LEVEL FEATURE REUSE

Figure 10 presents a comparative analysis of feature variations during the video generation process for the two distinct prompts introduced in Figure 4(a). Figure 10(a) illustrates the block-level feature differences, while Figure 10(b) depicts the timestep-level feature differences. The results reveal that block-level features exhibit significant and discernible variations between the two prompts. Specifically, the video generated for the more static scene (e.g., Prompt 1) demonstrates lower feature variability, enabling more granular and aggressive caching and reuse of block features. This facilitates substantial acceleration while maintaining visual fidelity. In contrast, the timestep-level features show minimal differentiation across prompts, making it difficult to perform prompt-specific adaptive caching control. This fundamental limitation explains why timestep-level caching methods, such as TeaCache, often result in inferior video quality compared to our proposed block-wise method.

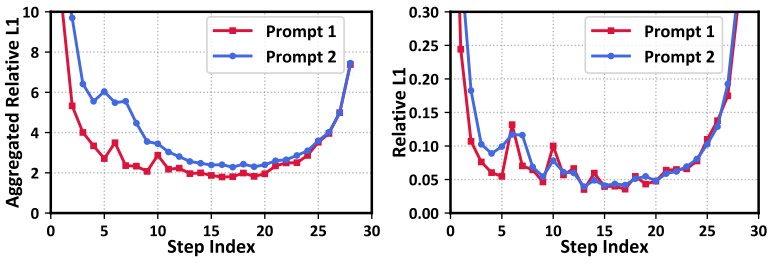

(a) Block-level caching performance   (b) Timestep-level caching performance

Figure 10: Comparative analysis of feature variations: Block-level vs. Timestep-level.

## B   EXPERIMENTAL SETTINGS

**Base Model.** This paper focuses on DiT-based video generation, evaluating five state-of-the-art open-source models: Open-Sora Zheng et al. (2024), Open-Sora-Plan Lin et al. (2024a), Latte Ma et al. (2025a), Wan 2.1 Wan et al. (2025), and HunyuanVideo Kong et al. (2024). Open-Sora employs Spatial-Temporal DiT Ho et al. (2022), allowing for the creation of videos up to 16 seconds long at 720p resolution. Open-Sora Plan focuses on generating high-resolution videos with extended durations based on diverse user inputs, featuring components like a Wavelet-Flow Variational Autoencoder and specialized denoisers. Latte effectively extracts spatio-temporal tokens from videos and utilizes DiT blocks to model the token distribution in a latent space. Wan is a comprehensive and open-source suite of high-performance video foundation models. HunyuanVideo is an open-source video foundation model, integrating advanced data curation and a scalable transformer architecture. The inference configs of five models are shown in Table 6, which strictly follow the official settings.

| Model | Scheduler | Steps | Text Encoder |
|---|---|---|---|
| Open-Sora | RFLOW | 30 | T5 |
| Open-Sora-Plan | PNDM | 150 | T5 |
| Latte | DDIM | 50 | T5 |
| Wan 2.1 | FlowMatch | 50 | UMT5 |
| HunyuanVideo | FlowMatch | 50 | MLLM |

Table 6: The inference config of base models.

## C   METRICS

**Peak Signal-to-Noise Ratio (PSNR).** PSNR is a common metric used to measure the quality of reconstruction in lossy compression and signal processing. It compares the maximum possible power of a signal to the power of corrupting noise. Higher PSNR values indicate better quality. The PSNR

is calculated as follows:

$$\text{PSNR} = 10 \cdot \log_{10}\left(\frac{R^2}{\text{MSE}}\right) \tag{9}$$

where $R$ is the maximum possible pixel value of the image and MSE denotes the Mean Squared Error between the reference image and the reconstructed image.

**Learned Perceptual Image Patch Similarity (LPIPS).** LPIPS Zhang et al. (2018) is a modern perceptual similarity metric that is based on deep learning. It quantifies the similarity between two images by comparing their features at different layers of a neural network. This metric is believed to better correlate with human visual perception than traditional metrics by capturing perceptually important features. The LPIPS score between two images $I$ and $K$ is typically computed as:

$$\text{LPIPS}(I, K) = \frac{1}{L}\sum_{l=1}^{L}\|\phi_l(I) - \phi_l(K)\|^2 \tag{10}$$

where $L$ is the number of layers in the pretrained neural network, and $\phi_l$ denotes the feature extraction at layer $l$. For video evaluation, LPIPS is calculated for each frame of the video and averaged across all frames to produce a final score.

**Structural Similarity Index Measure (SSIM).** SSIM Wang & Bovik (2002) is a perceptual metric that assesses the visual impact of three characteristics: luminance, contrast, and structure. Instead of measuring pixel-wise differences, SSIM compares the structural information in the images, making it a more relevant measure of perceived quality. The SSIM index is calculated as follows:

$$SSIM(I, K) = \frac{(2\mu_I\mu_K + C_1)(2\sigma_{IK} + C_2)}{(\mu_I^2 + \mu_K^2 + C_1)(\sigma_I^2 + \sigma_K^2 + C_2)} \tag{11}$$

where $\mu_I$ and $\mu_K$ are the average luminance of $I$ and $K$, $\sigma_I^2$ and $\sigma_K^2$ are the variances of $I$ and $K$, $\sigma_{IK}$ is the covariance between $I$ and $K$, $C_1$ and $C_2$ are constants used to stabilize the division with weak denominator. For video evaluation, SSIM is calculated for each frame and then averaged over all frames to provide an overall similarity measure.

## D  ADDITIONAL EXPERIMENTAL RESULTS

### D.1  COMPARISON WITH HIGH-MEMORY METHODS ON A SINGLE GPU.

We compare BWCache with high-memory caching methods, including ProfilingDiT Ma et al. (2025b) and TaylorSeer Liu et al. (2025b). These methods require caching extensive intermediate features, leading to significant memory overhead. When generating long videos, both ProfilingDiT and TaylorSeer encounter out-of-memory (OOM) errors. To ensure a fair comparison, we evaluate these methods on shorter video sequences where they can run successfully. Specifically, we test on Wan 2.1 with 9 frames at 480P and HunyuanVideo with 17 frames at 544P, which are the maximum video lengths that TaylorSeer can process on a single NVIDIA A800 80GB GPU, as shown in Table 7. We configure ProfilingDiT with $T_s = 6$, and TaylorSeer with $\mathcal{O} = 1$ and $\mathcal{N} = 3$ for Wan 2.1, and $\mathcal{O} = 1$ and $\mathcal{N} = 4$ for HunyuanVideo. Despite operating under more favorable conditions for the baseline methods, BWCache still achieves superior visual quality metrics while maintaining competitive or better acceleration performance. This demonstrates that BWCache provides a better balance between memory efficiency and generation quality compared to methods that require extensive feature caching.

### D.2  FULL VBENCH RESULTS.

VBench Huang et al. (2024b) is a comprehensive benchmark suite designed for evaluating video generative models. It provides a standardized framework that includes diverse evaluation metrics, datasets, and protocols to facilitate in-depth comparisons of model performance across various tasks. This systematic approach ensures reliable assessments of generated videos in terms of quality, efficiency, and robustness. Table 8 compares the performance of four video generation methods across each of the 16 VBench dimensions. A higher score indicates relatively better performance for a particular dimension. We also provide two specially built baselines, i.e., Empirical Min and Max (the approximated achievable min and max scores for each dimension), as references.

| Model | Method | Visual Quality | | | | Efficiency | |
|---|---|---|---|---|---|---|---|
| | | VBench↑ | LPIPS↓ | SSIM↑ | PSNR↑ | Speedup↑ | Latency(s)↓ |
| **Wan 2.1** (9 frames, 480P) | Original | 77.05% | - | - | - | 1× | 287 |
| | ProfilingDiT | 76.53% | 0.2053 | 0.8221 | 24.20 | 1.24× | 231 |
| | TaylorSeer | 75.13% | 0.3080 | 0.7231 | 18.75 | **1.87×** | **153** |
| | BWCache | **76.81%** | **0.0929** | **0.9048** | **29.65** | 1.67× | 171 |
| **HunyuanVideo** (17 frames, 544P) | Original | 82.08% | - | - | - | 1× | 179 |
| | ProfilingDiT | 80.33% | 0.2610 | 0.6662 | 18.77 | 1.72× | 104 |
| | TaylorSeer | **82.05%** | 0.4021 | 0.5526 | 15.33 | 1.94× | 92 |
| | BWCache | 81.42% | **0.1804** | **0.7507** | **21.60** | **2.39×** | **75** |

Table 7: Comparison with high-memory caching methods on reduced video lengths.

| Models | Subject Consistency | Background Consistency | Temporal Flickering | Motion Smoothness | Dynamic Degree | Aesthetic Quality | Imaging Quality | Object Class |
|---|---|---|---|---|---|---|---|---|
| Original | 95.08% | 96.86% | 99.41% | 98.80% | 42.00% | 59.53% | 62.60% | 95.87% |
| PAB | 95.15% | 96.70% | 99.38% | 99.09% | 28.00% | 56.32% | 59.63% | 96.37% |
| TeaCache | 95.33% | 96.83% | 99.44% | 98.94% | 36.00% | 58.50% | 59.79% | 96.00% |
| BWCache | 95.91% | 96.76% | 99.43% | 98.97% | 38.00% | 59.26% | 61.29% | 98.25% |
| Empirical Min | 14.62% | 26.15% | 62.93% | 70.60% | 0.00% | 0.00% | 0.00% | 0.00% |
| Empirical Max | 100.00% | 100.00% | 100.00% | 99.75% | 100.00% | 100.00% | 100.00% | 100.00% |

| Models | Multiple Objects | Human Action | Color | Spatial Relationship | Scene | Appearance Style | Temporal Style | Overall Consistency |
|---|---|---|---|---|---|---|---|---|
| Original | 36.00% | 84.00% | 81.25% | 85.75% | 59.62% | 25.40% | 23.03% | 27.90% |
| PAB | 38.50% | 68.00% | 85.64% | 77.22% | 57.37% | 25.06% | 22.20% | 26.70% |
| TeaCache | 31.50% | 82.00% | 81.61% | 85.75% | 56.13% | 24.93% | 22.73% | 27.24% |
| BWCache | 36.25% | 84.00% | 82.58% | 86.52% | 59.25% | 25.27% | 22.74% | 27.91% |
| Empirical Min | 0.00% | 0.00% | 0.00% | 0.00% | 0.00% | 0.00% | 0.00% | 0.00% |
| Empirical Max | 100.00% | 100.00% | 100.00% | 100.00% | 82.22% | 28.55% | 36.40% | 36.40% |

Table 8: VBench evaluation results per dimension.

## D.3 LAST STEP

In the final stage of the DiT model for video generation, that is, during the last stage of denoising, the reuse of cached block features can introduce unnecessary biases and lead to a degradation in the generated video. Therefore, assuming that cache reuse is triggered at the $k$-th step during the denoising process, we default that the last $1/2k$ steps do not involve cache and reuse operations. Meanwhile, we have statistically analyzed the results for the last $1/3k$ and $2/3k$ steps, as well as fixed 5 and 8 steps, as shown in Table 9. It can be seen that the choice of the last steps has a significant impact on latency and video generation quality. Specifically, as the number of last steps increases, latency tends to decrease, while metrics exhibit fluctuations to some extent. In general, our default setting $1/2k$ balances the generation quality and latency.

| Last Step | Latency(s)↓ | Vbench↑ | LPIPS↓ | SSIM↑ | PSNR↑ |
|---|---|---|---|---|---|
| $1/3k$ | 29.10 | 79.95% | 0.0922 | 0.8823 | 26.95 |
| $1/2k$ | 27.68 | 80.03% | 0.0879 | 0.8854 | 27.05 |
| $2/3k$ | 25.98 | 80.09% | 0.0853 | 0.8874 | 27.16 |
| 5 | 31.47 | 80.13% | 0.0877 | 0.8866 | 27.10 |
| 8 | 28.62 | 80.00% | 0.0862 | 0.8896 | 27.01 |

Table 9: Impact of different last steps.

## D.4 STATISTICAL ANALYSIS

Table 10 shows the statistical results of the paired t-test that determine the significance level of the BWCache compared with other methods regarding LPIPS, SSIM, and PSNR. Specifically, the table provides the mean and standard deviation of PAB Zhao et al. (2024b) and TeaCache Liu et al. (2024), alongside the p-value and corresponding confidence interval. Our null hypothesis is: "There is no significant difference in the performance between the BWCache and other methods". The table shows a meaningful difference between the BWCache and other methods, while the p-value is lower than 0.05 in all cases. There is statistically significant evidence at the 5% significance level

| Model | Method | Metrics | Mean | Standard Deviation | P-Value | Confidence Interval |
|---|---|---|---|---|---|---|
| **Open-Sora** | PAB | LPIPS | 0.2134 | 0.1026 | 9.90e-111 | [0.2048, 0.2220] |
| | | SSIM | 0.7798 | 0.1084 | 8.99e-73 | [0.7707, 0.7888] |
| | | PSNR | 22.0191 | 2.8325 | 2.8e-135 | [21.7819, 22.2564] |
| | TeaCache | LPIPS | 0.1496 | 0.0780 | 1.07e-46 | [0.1430, 0.1561] |
| | | SSIM | 0.8104 | 0.1034 | 1.09e-42 | [0.8017, 0.8190] |
| | | PSNR | 22.3929 | 4.3654 | 2.8e-80 | [22.0273, 22.7586] |
| | BWCache | LPIPS | 0.0879 | 0.0561 | - | [0.0832, 0.0926] |
| | | SSIM | 0.8854 | 0.0665 | - | [0.8798, 0.8909] |
| | | PSNR | 27.0510 | 2.9898 | - | [26.8006, 27.3014] |
| **Open-Sora-Plan** | PAB | LPIPS | 0.2781 | 0.1177 | 7.69e-51 | [0.2602, 0.2961] |
| | | SSIM | 0.6627 | 0.1449 | 7.16e-37 | [0.6407, 0.6848] |
| | | PSNR | 19.4968 | 3.6949 | 6.76e-42 | [18.9340, 20.0596] |
| | TeaCache | LPIPS | 0.1965 | 0.0928 | 6.64e-27 | [0.1824, 0.2106] |
| | | SSIM | 0.7502 | 0.1090 | 4.95e-18 | [0.7336, 0.7668] |
| | | PSNR | 21.5010 | 3.5785 | 6.25e-24 | [20.9559, 22.0461] |
| | BWCache | LPIPS | 0.1001 | 0.0520 | - | [0.0922, 0.1080] |
| | | SSIM | 0.8435 | 0.0743 | - | [0.8322, 0.8548] |
| | | PSNR | 25.8734 | 3.7611 | - | [25.3006, 26.4463] |
| **Latte** | PAB | LPIPS | 0.4625 | 0.1481 | 3.84e-218 | [0.4501, 0.4750] |
| | | SSIM | 0.5964 | 0.1631 | 3.88e-106 | [0.5827, 0.6100] |
| | | PSNR | 17.0299 | 2.2196 | 2.67e-189 | [16.8440, 17.2158] |
| | TeaCache | LPIPS | 0.1969 | 0.1058 | 5.32e-17 | [0.1880, 0.2058] |
| | | SSIM | 0.7606 | 0.1329 | 2.39e-12 | [0.7495, 0.7718] |
| | | PSNR | 22.1947 | 4.3783 | 1.02e-40 | [21.8280, 22.5614] |
| | BWCache | LPIPS | 0.1399 | 0.1160 | - | [0.1301, 0.1496] |
| | | SSIM | 0.8181 | 0.1357 | - | [0.8067, 0.8294] |
| | | PSNR | 26.4620 | 5.6981 | - | [25.9847, 26.9392] |
| **Wan 2.1** | PAB | LPIPS | 0.1623 | 0.1141 | 2.10e-18 | [0.1440, 0.1806] |
| | | SSIM | 0.7643 | 0.1465 | 4.92e-09 | [0.7405, 0.7881] |
| | | PSNR | 21.3210 | 3.5476 | 3.07e-22 | [20.7974, 21.8446] |
| | TeaCache | LPIPS | 0.2382 | 0.0894 | 1.49e-50 | [0.2232, 0.2531] |
| | | SSIM | 0.6608 | 0.1406 | 1.47e-31 | [0.6373, 0.6843] |
| | | PSNR | 18.6197 | 3.2942 | 1.11e-42 | [18.0692, 19.1702] |
| | BWCache | LPIPS | 0.0805 | 0.0461 | - | [0.0728, 0.0882] |
| | | SSIM | 0.8527 | 0.0968 | - | [0.8365, 0.8689] |
| | | PSNR | 25.7246 | 3.9350 | - | [25.0670, 26.3821] |
| **HunyuanVideo** | PAB | LPIPS | 0.1045 | 0.1224 | 3.20e-06 | [0.0857, 0.1233] |
| | | SSIM | 0.8341 | 0.1642 | 1.47e-03 | [0.8070, 0.8613] |
| | | PSNR | 26.6958 | 3.6582 | 2.11e-11 | [26.1089, 27.2827] |
| | TeaCache | LPIPS | 0.1685 | 0.0823 | 3.68e-25 | [0.1547, 0.1822] |
| | | SSIM | 0.7973 | 0.1184 | 2.80e-13 | [0.7775, 0.8171] |
| | | PSNR | 23.9004 | 3.9613 | 1.14e-25 | [23.2384, 24.5623] |
| | BWCache | LPIPS | 0.0768 | 0.0468 | - | [0.0689, 0.0846] |
| | | SSIM | 0.8922 | 0.0860 | - | [0.8778, 0.9065] |
| | | PSNR | 29.6333 | 4.2949 | - | [28.9157, 30.3510] |

Table 10: Statistical comparison of the BWCache with compared methods.

to suggest that the BWCache performs better than PAB and TeaCache. Thus, the null hypothesis is rejected, which proves that the differences are significant.

## D.5 MEMORY USAGE ANALYSIS

As shown in Table 11, we compare the memory consumption of the original models, our proposed BWCache method, and all main baseline methods (PAB, TeaCache, FasterCache, ProfilingDiT, and TaylorSeer) across five models under various video generation configurations. The results indicate that all training-free caching methods require additional GPU memory compared to the original models due to caching intermediate features from previous timesteps. This memory overhead is inherent to all caching-based acceleration approaches. For models already at absolute memory limits, all training-free caching methods face similar constraints. Notably, Wan 2.1 and HunyuanVideo employ full 3D self-attention Peebles & Xie (2023), which inherently demand more memory. For instance, TaylorSeer can only process Wan 2.1 with 9 frames at 480P and HunyuanVideo with 17 frames at 544P, which are the maximum video lengths that TaylorSeer can process on a single NVIDIA A800 80GB GPU. Compared to other cache-based methods, BWCache demonstrates competitive or superior memory efficiency.

Moreover, BWCache is not an all-or-nothing design; it can be configured to reduce memory overhead for memory-constrained settings. The method can be configured to: (1) only cache a subset of

blocks instead of all blocks, (2) apply pooling or compression to cached features, or (3) when only caching the first DiT block each timestep, BWCache becomes similar to TeaCache. To maintain computational efficiency, BWCache selectively uses a subset of blocks for cache indicator calculation in Wan 2.1 and HunyuanVideo models. Table 12 investigates the effect of varying the proportion of blocks used for cache indicator computation on video generation quality. We compute the similarity indicator using a uniformly spaced subset of blocks across depth. This design aims to cover both early, middle, and late blocks while reducing the amount of cached state and indicator computation. Generally, a higher proportion of blocks leads to more accurate cache indicator calculation. To balance latency and efficiency, we default to using 50% of the blocks for cache indicator calculation in these models. Although BWCache introduces additional memory usage, it significantly accelerates inference and reduces overall GPU operational costs, making it a highly effective solution for practical applications.

| Model | Open-Sora | Open-Sora-Plan | Latte |
|---|---|---|---|
| **Parameter** | **51 frames, 480P** | **65 frames, 512 × 512** | **16 frames, 512 × 512** |
| Original | 15270MiB | 15724MiB | 15240MiB |
| PAB | 27016MiB | 26722MiB | 29126MiB |
| TeaCache | 20630MiB | 21266MiB | 18499MiB |
| FasterCache | 26130MiB | 22508MiB | 23264MiB |
| BWCache | 21362MiB | 18022MiB | 17400MiB |

| Model | Wan 2.1 | | |
|---|---|---|---|
| **Parameter** | **9 frames, 480P** | **81 frames, 480P** | **81 frames, 720P** |
| Original | 49757MiB | 71861MiB | 73955MiB |
| PAB | 59757MiB | 80285MiB | 80154MiB |
| TeaCache | 56737MiB | 74197MiB | 76563MiB |
| ProfilingDiT | 57089MiB | 78787MiB | OOM |
| TaylorSeer | 79263MiB | OOM | OOM |
| BWCache | 58047MiB | 79541MiB | 80607MiB |

| Model | HunyuanVideo | | |
|---|---|---|---|
| **Parameter** | **17 frames, 720P** | **129 frames, 540P** | **129 frames, 720P** |
| Original | 45124MiB | 58491MiB | 74479MiB |
| PAB | 57411MiB | 78787MiB | OOM |
| TeaCache | 52451MiB | 64729MiB | 78039MiB |
| ProfilingDiT | 65124MiB | OOM | OOM |
| TaylorSeer | 74287MiB | OOM | OOM |
| BWCache | 49757MiB | 66184MiB | 80625MiB |

Table 11: Memory usage comparison of BWCache and baseline methods.

| Wan 2.1 (81 frames, 480P) | Memory usage | LPIPS↓ | SSIM↑ | PSNR↑ |
|---|---|---|---|---|
| 10% blocks | 72367MiB | 0.0790 | 0.8524 | 25.79 |
| 30% blocks | 75621MiB | 0.0789 | 0.8533 | 25.83 |
| 50% blocks | 79541MiB | 0.0782 | 0.8539 | 25.86 |
| **HunyuanVideo (129 frames, 540P)** | **Memory usage** | **LPIPS↓** | **SSIM↑** | **PSNR↑** |
| 10% blocks | 60977MiB | 0.0795 | 0.8921 | 29.95 |
| 30% blocks | 63617MiB | 0.0786 | 0.8884 | 29.86 |
| 50% blocks | 66184MiB | 0.0794 | 0.8903 | 29.91 |

Table 12: Impact of different block proportions on cache indicator calculation.

## D.6 COMPUTATIONAL LATENCY OF THE CACHE INDICATOR

As shown in Table 13, we measure the absolute time spent on computing the cache indicator and its relative proportion of the total inference time across five models. The percentages in parentheses indicate the fraction of the entire inference process consumed by the cache indicator computation. The results demonstrate that the computational cost of the cache indicator is relatively modest, ranging from 10.6% to 22.5% of the total inference time. Despite this overhead, BWCache still achieves

significant overall speedup because the cache indicator computation is substantially faster than re-computing all DiT blocks, and the cached features enable skipping a large portion of redundant computations.

| Model | Open-Sora | Open-Sora-Plan | Latte | Wan 2.1 | HunyuanVideo |
|---|---|---|---|---|---|
| Computational latency (s) | 6.22(22.5%) | 5.05(11.4%) | 3.12(22.0%) | 48.23(10.6%) | 68.90(15.9%) |

Table 13: Computational latency of the cache indicator.

## D.7 DYNAMIC VIDEO GENERATION

To thoroughly evaluate the performance of BWCache in highly dynamic scenarios, we specifically analyze video generation tasks involving human motion, fast parallax, and other compositional scenes with high motion complexity. While our main experiments have already included various dynamic scenes, we further extract and evaluate a subset of videos that exhibit high dynamic characteristics. As shown in Table 14 and Figure 11, we evaluate BWCache's performance on highly dynamic video generation tasks across five models. VBench is a comprehensive benchmarking suite that enables quantitative assessment of how well our method handles challenging scenarios with rapid motion and complex scene compositions. The results demonstrate that BWCache maintains competitive visual quality even in highly dynamic scenarios, with only minimal quality degradation compared to the original models. This indicates that the block-wise adaptive caching mechanism effectively adapts to varying scene dynamics.

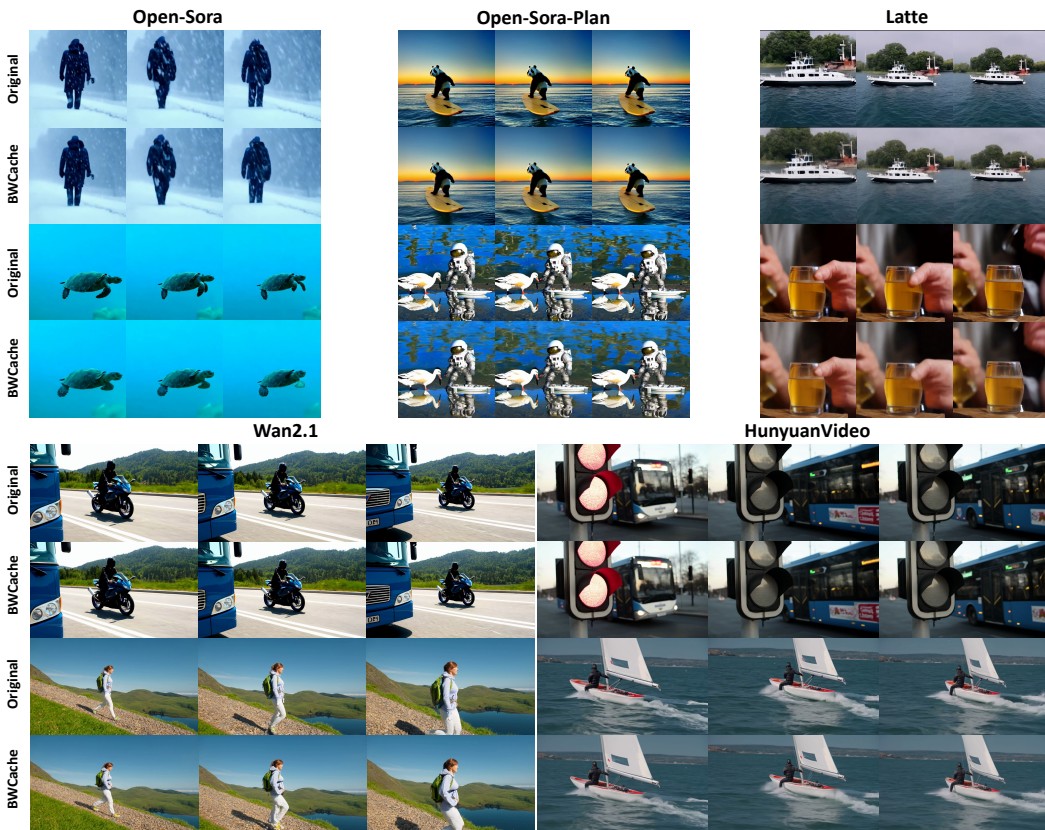

Figure 11: Dynamic video generation.

| Model | Open-Sora | Open-Sora-Plan | Latte | Wan 2.1 | HunyuanVideo |
|---|---|---|---|---|---|
| Original | 80.12% | 80.48% | 78.23% | 82.03% | 82.14% |
| BWCache | 79.71% | 80.37% | 77.64% | 81.90% | 81.38% |

Table 14: Dynamic video generation results.

| Model | Open-Sora | Open-Sora-Plan | Latte | Wan 2.1 | HunyuanVideo |
|---|---|---|---|---|---|
| **Threshold** | 0.15 - 0.25 | 0.15 - 0.25 | 0.1 - 0.2 | 0.1 - 0.2 | 0.15 - 0.25 |
| **VBench↑** | 79.3% - 80.0% | 80.2% - 81.0% | 75.5% - 78.7% | 80.4% - 81.3% | 81.9% - 82.6% |
| **Speedup↑** | 1.41 - 1.96 $\times$ | 2.02 - 2.52 $\times$ | 1.67 - 2.07 $\times$ | 1.75 - 2.48 $\times$ | 2.31 - 2.94 $\times$ |

Table 15: Recommended parameter settings.

# E DISCUSSION

## E.1 GUIDELINES FOR PARAMETERS SETTING

When applying BWCache to a new DiT-based model, we recommend starting with a default value of $\delta = 0.15$, which prioritizes quality. To further optimize, generate a few sample videos and plot the ARL1 across timesteps to identify the optimal region for cache reuse, then iteratively test different threshold values to balance quality and speedup. Models with more stable feature dynamics may tolerate higher thresholds, while those handling highly dynamic content may require lower thresholds. During threshold selection, prioritize maintaining VBench scores within 1-2% of the original model's performance, and reduce the threshold if quality degradation exceeds acceptable limits.

In practice, most users can directly use the default value of $\delta = 0.15$ or select from the validated ranges in Table 15 based on their model type, as these have been extensively tested across diverse models and generation tasks.

For the reuse interval $R$, we recommend setting it to 10% of the total timesteps as the default value. Our extensive experiments across all five models demonstrate that a reuse interval of 10% consistently achieves an optimal balance between inference efficiency and visual quality.

## E.2 FEATURE DIFFERENCE PATTERN IN OTHER SETTINGS

We also investigate whether similar patterns exist in other diffusion model settings. Specifically, as shown in Figure 12 and 13, class-conditional generation and image generation exhibit an inverted L-shaped pattern, which differs from the U-shaped pattern observed in video generation models. The feature variation pattern is closely related to the noise schedule and loss function used during training Zhang et al. (2024b). In text-to-image generation, the model rapidly establishes the overall semantic structure and content layout, leading to high feature variation. Once the semantic structure is established, the remaining timesteps focus primarily on refining details and improving visual quality, resulting in stable and gradually decreasing feature variations.

## E.3 LATENT DRIFT FROM FEATURE REUSE

While the similarity indicator effectively identifies when features are stable enough for cache reuse, we observe that once cache reuse begins, the subsequent feature changes become unpredictable. As illustrated in Figure 14, after multiple cycles of cache reuse, the feature variations exhibit irregular patterns that deviate from the expected U-shaped trajectory. In this case, each reuse step propagates small discrepancies, which accumulate over time and cause the latent representation to drift away from the intended denoising path. This latent drift introduces additional error that cannot be easily predicted or corrected by dynamically monitoring the L1 distance alone.

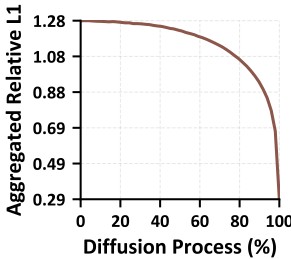

Figure 12: Feature changes for text-to-image generation.

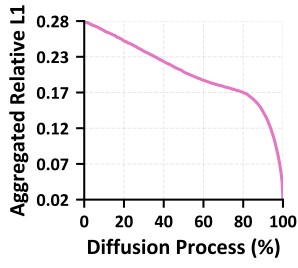

Figure 13: Feature changes for class-conditional generation.

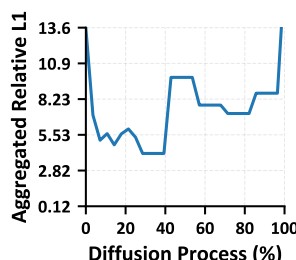

Figure 14: Feature changes during cache reuse.

## F    MORE VISUAL RESULTS

The additional visual comparison results for Open-Sora, Open-Sora-Plan, Latte, Wan 2.1, and Hun-yuanVideo are presented in Figure 15, Figure 16, Figure 17, Figure 18, and Figure 19. All prompts sourced from the Open-Sora gallery Lin et al. (2024a), VBench benchmark Huang et al. (2024b), and T2V-CompBench Sun et al. (2024), which collectively offer diverse and representative scenarios for comprehensive evaluation. Our method demonstrates consistent fidelity across diverse models, styles, and content types in video generation while maintaining computational efficiency.

## STATEMENT ON LARGE LANGUAGE MODEL (LLM) USAGE

In the preparation of this manuscript, large language models (LLMs) were utilized solely for the purpose of language polishing. The LLM played no role in the generation of core ideas. All final content was thoroughly reviewed and approved by the authors.

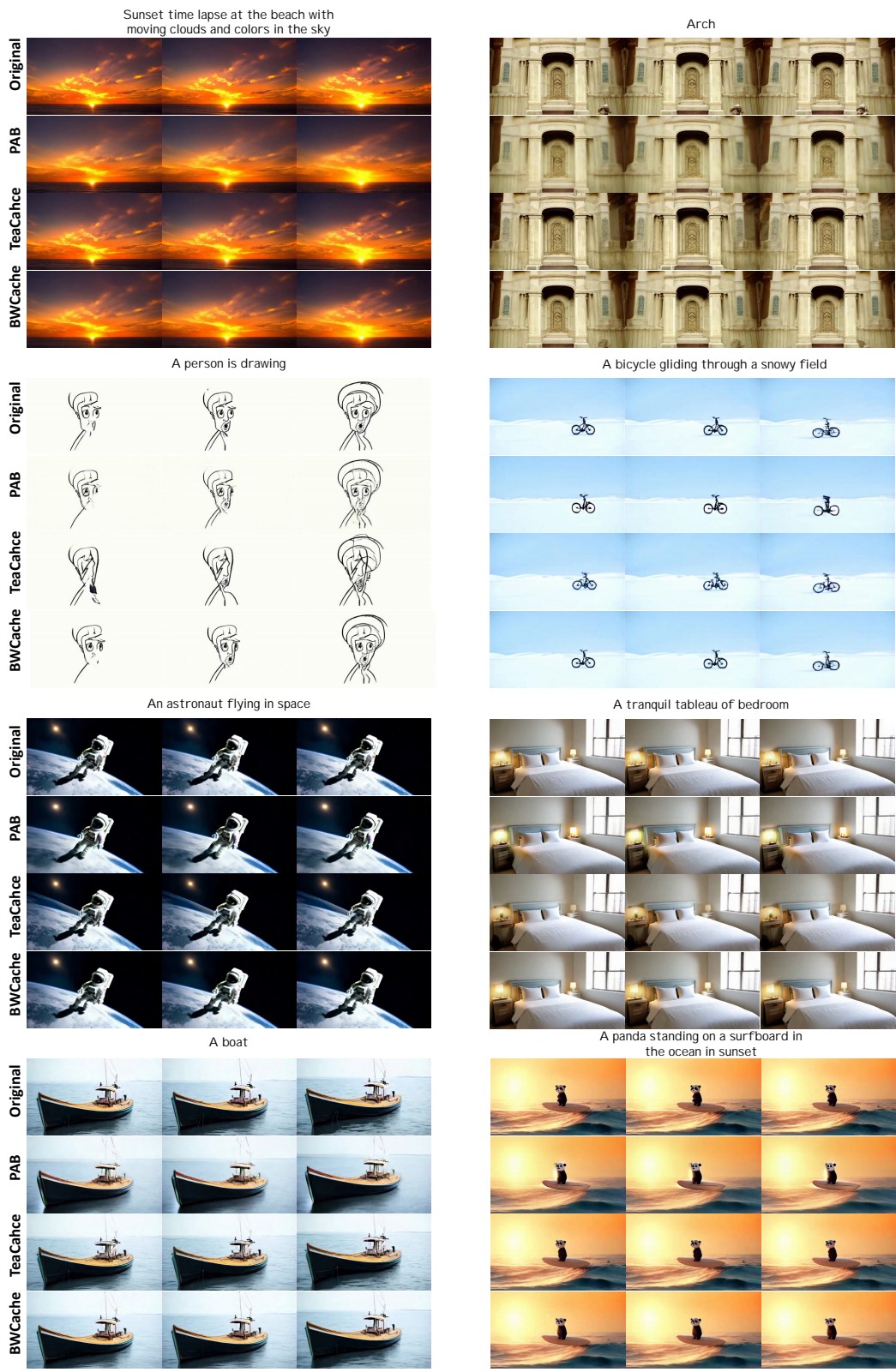

Figure 15: More visual results on Open-Sora (51 frames, 480P).

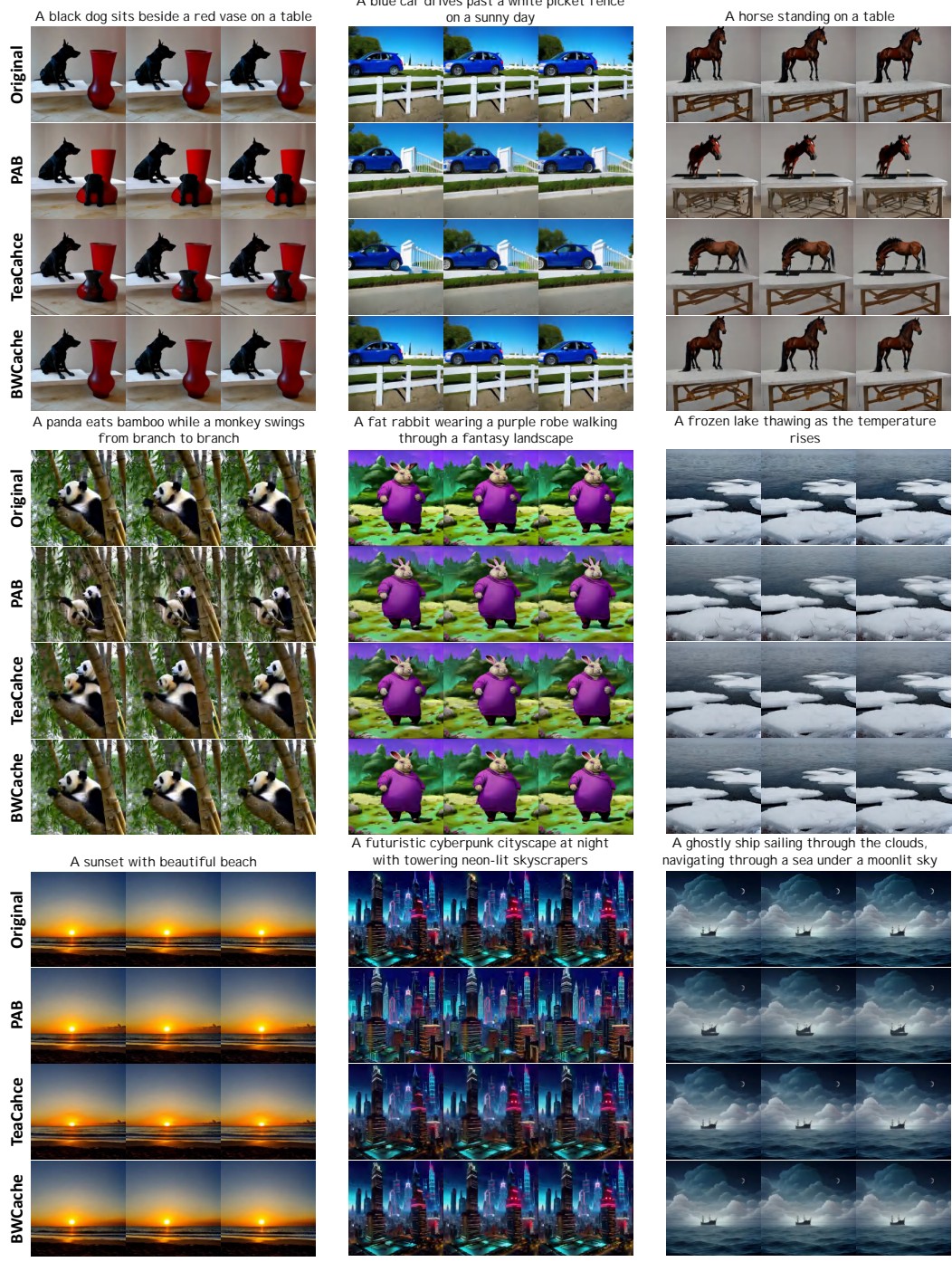

Figure 16: More visual results on Open-Sora-Plan (65 frames, 512×512).

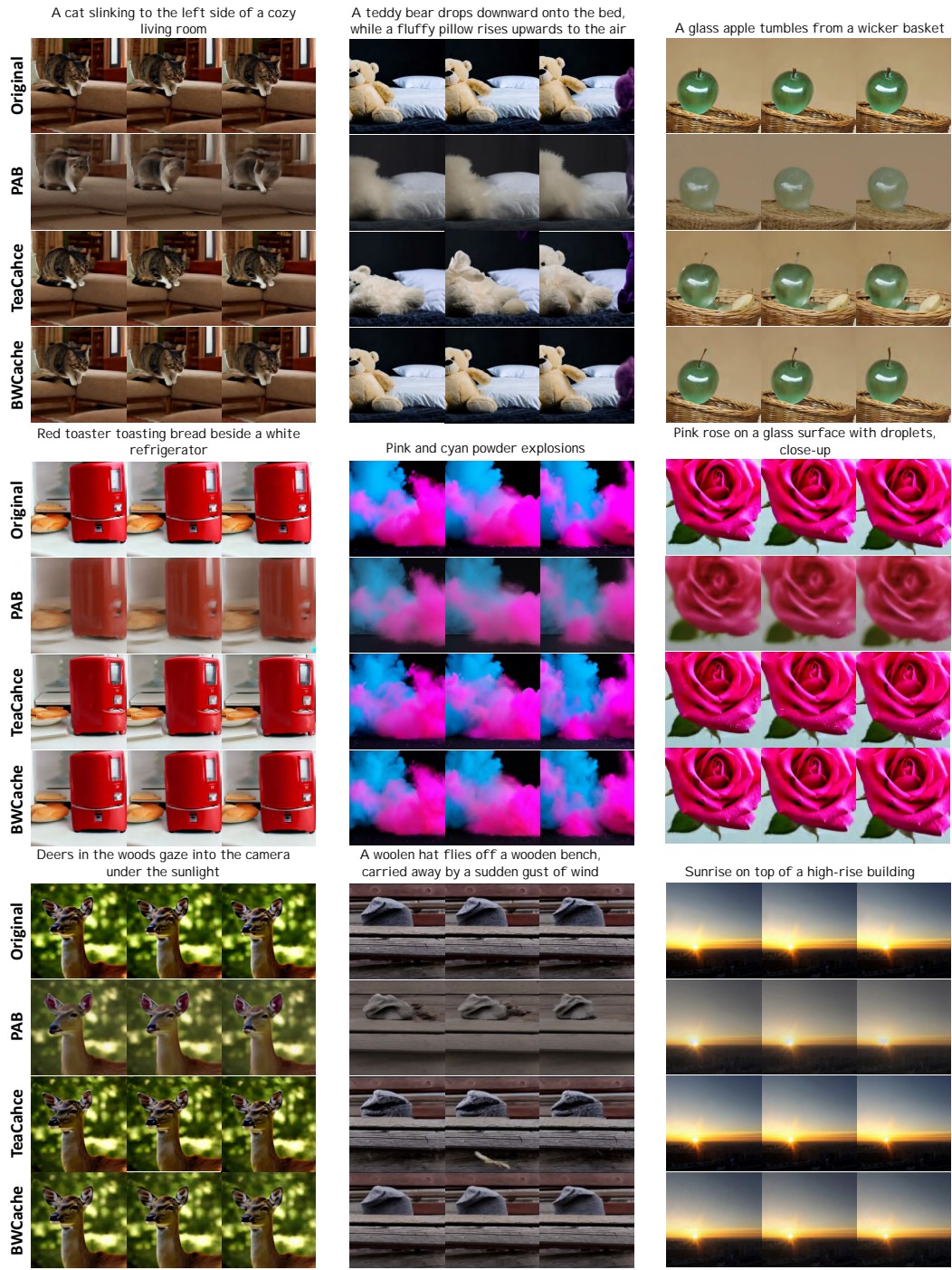

Figure 17: More visual results on Latte (16 frames, 512×512).

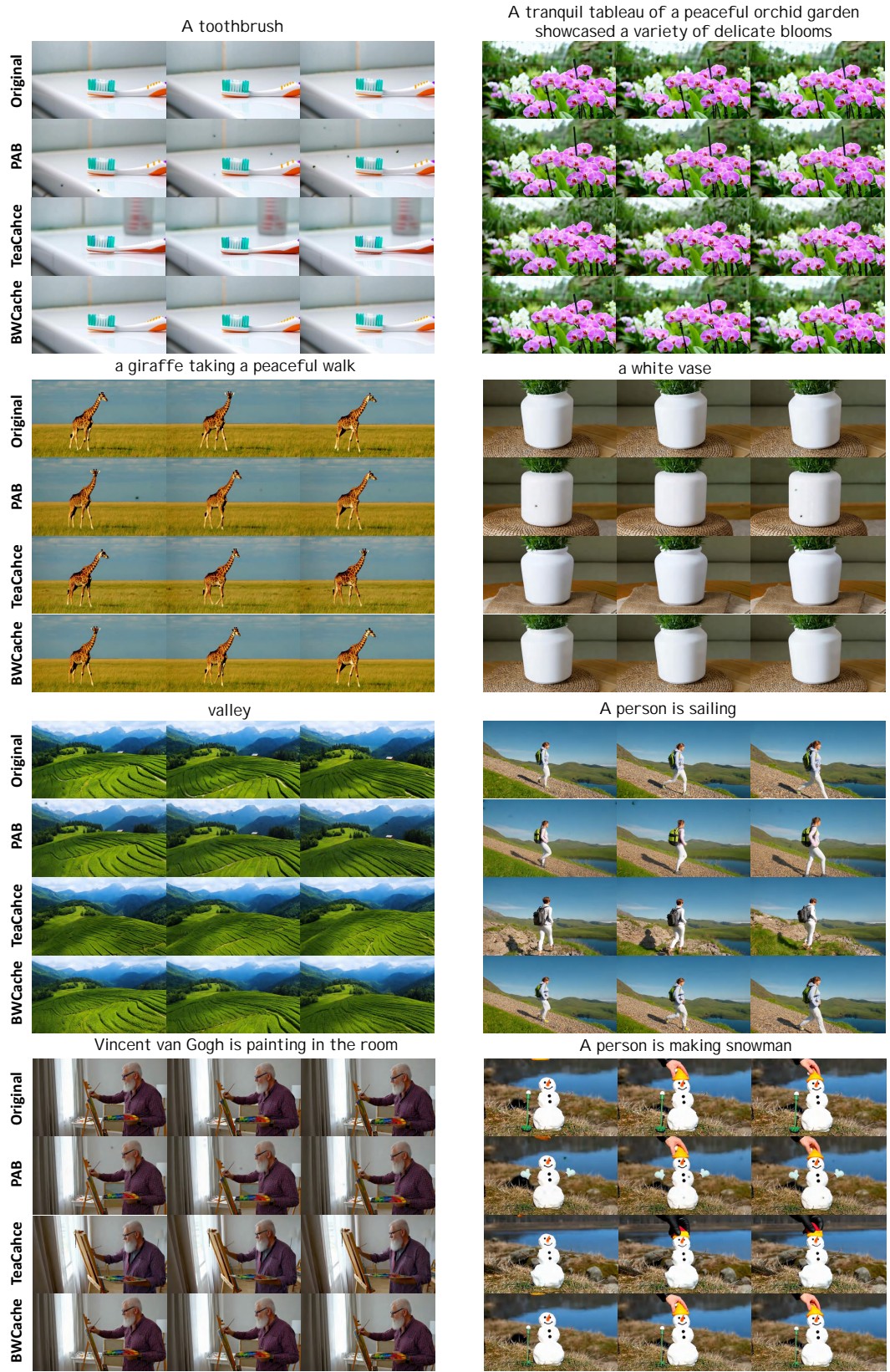

Figure 18: More visual results on Wan 2.1 (81 frames, 480P).

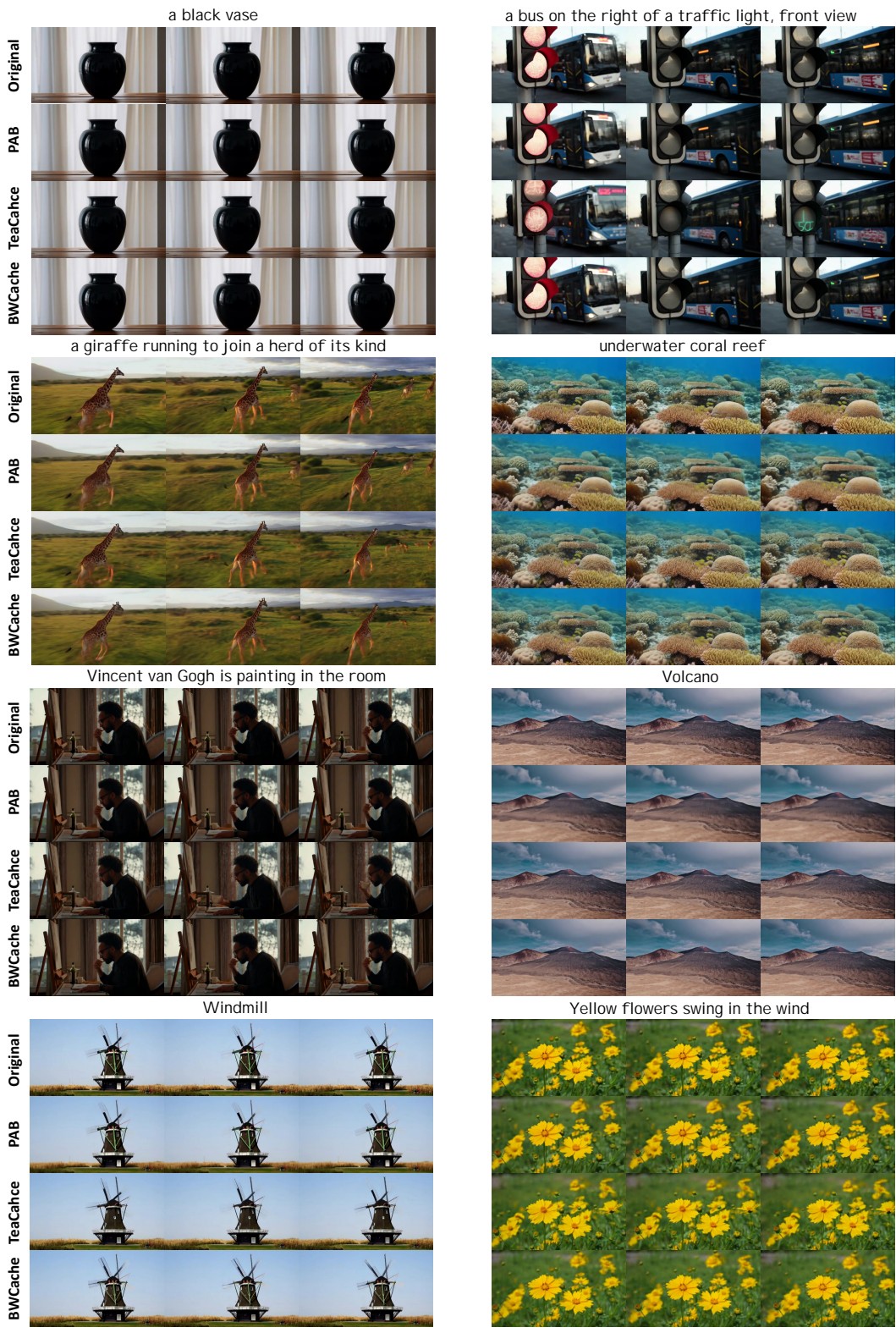

Figure 19: More visual results on HunyuanVideo (129 frames, 544P).

