# OpenReview forum: "BWCache: Accelerating Video Diffusion Transformers through Block-Wise Caching"
_ICLR.cc/2026/Conference — ICLR 2026 Poster_

### Official Review · Reviewer_wT4r · 2025-10-26

**Soundness:** 2
**Presentation:** 2
**Contribution:** 3
**Rating:** 6
**Confidence:** 4

**Summary:**

The paper proposes BWCache, a training-free acceleration method for DiT-based video diffusion. It caches and reuses block-level features across timesteps and triggers reuse with a similarity indicator computed from the mean relative L1 distance between adjacent-timestep block features. To avoid latent drift, it adds periodic recomputation with a reuse interval. Experiments on Open-Sora, Open-Sora-Plan, Latte, Wan 2.1, and HunyuanVideo report up to 2.6× speedup with comparable visual quality, along with scaling results under multi-GPU DSP and ablations on the indicator threshold and reuse interval.

**Strengths:**

1. The mechanism is training-free, compatible with a range of DiT video generators, and easy to integrate at inference time.

2. The use of a block-wise mean relative L1 indicator to decide reuse is principled and aligns with the observed per-timestep dynamics.

3. Results span multiple DiT-based models and include comparisons to ∆-DiT, T-GATE, PAB, and TeaCache, plus scaling behavior with DSP and ablations over threshold δ and reuse interval R.

4. The paper gives quantitative and qualitative evidence that BWCache sustains visual fidelity at meaningful acceleration levels.

**Weaknesses:**

1. The comparison with related work in Table 1 is not sufficiently comprehensive. Several recent and relevant caching-based acceleration methods, such as FasterCache and TaylorSeer, are missing from the comparison.

2. The appendix shows that the proposed method increases memory consumption compared with the original base model. It should also include the memory usage of other representative cache-based methods to demonstrate the superiority of the proposed approach.

**Questions:**

See Weakness

---

> ### Author Response · Authors · 2025-11-23
> **Response to Reviewer wT4r**
>
> **_Q1: The comparison with related work in Table 1 is not sufficiently comprehensive. Several recent and relevant caching-based acceleration methods, such as FasterCache and TaylorSeer, are missing from the comparison._**
>
> **Response:** Thank you for your suggestion. We have now included FasterCache and TaylorSeer in our evaluation.
>
> **FasterCache Comparison:** The official FasterCache repository only provides support for Open-Sora, Open-Sora-Plan, and Latte models. The detailed experimental results are shown in Table 1:
>
>    | Model | Method | VBench↑ | LPIPS↓ | SSIM↑ | PSNR↑ | Speedup↑ | Latency(s)↓ |
>    |-------|--------|---------|--------|-------|-------|----------|-------------|
>    | Open-Sora (51 frames, 480P) | FasterCache | 79.21% | 0.1165 | 0.8435 | 23.99 | 1.35× | 32.03 |
>    | | BWCache | **80.03%** | **0.0879** | **0.8854** | **27.05** | **1.61×** | **27.68** |
>    | Open-Sora-Plan (65 frames, 512×512) | FasterCache | 80.61% | 0.1190 | 0.8103 | 24.06 | 1.51× | 65.79 |
>    | | BWCache | **80.82%** | **0.1001** | **0.8435** | **25.87** | **2.24×** | **44.49** |
>    | Latte (16 frames, 512×512) | FasterCache | 78.21% | 0.2442 | 0.7304 | 20.66 | 1.38× | 19.56 |
>    | | BWCache | **78.28%** | **0.1399** | **0.8181** | **26.46** | **1.90×** | **14.16** |
>
> **TaylorSeer Comparison:** The official TaylorSeer repository only provides support for Wan 2.1 and HunyuanVideo models. When generating long videos, TaylorSeer encounters out-of-memory (OOM) errors. To ensure a fair comparison, we evaluate these methods on shorter video sequences. We configure TaylorSeer with $\mathcal{O}=1$ and $\mathcal{N}=3$ for Wan 2.1, and $\mathcal{O}=1$ and $\mathcal{N}=4$ for HunyuanVideo. The detailed experimental results are shown in Table 7:
>
>    | Model | Method | VBench↑ | LPIPS↓ | SSIM↑ | PSNR↑ | Speedup↑ | Latency(s)↓ |
>    |-------|--------|---------|--------|-------|-------|----------|-------------|
>    | Wan 2.1 (9 frames, 480P) | TaylorSeer | 76.53% | 0.3080 | 0.7231 | 18.75 | **1.87×** | **153** |
>    | | BWCache | **76.81%** | **0.0929** | **0.9048** | **29.65** | 1.67× | 171 |
>    | HunyuanVideo (17 frames, 544P) | TaylorSeer | **82.05%** | 0.4021 | 0.5526 | 15.33 | 1.94× | 92 |
>    | | BWCache | 81.42% | **0.1804** | **0.7507** | **21.60** | **2.39×** | **75** |
>
> Overall, BWCache achieves a better balance of visual quality and efficiency compared to FasterCache and TaylorSeer.
>
> **_Q2: The appendix shows that the proposed method increases memory consumption compared with the original base model. It should also include the memory usage of other representative cache-based methods to demonstrate the superiority of the proposed approach._**
>
> **Response:** Thank you for your suggestion. We have expanded our memory usage analysis to include comparisons with all main baseline methods across different models and video generation configurations.
>
> The detailed memory usage results are shown in Table 11. Notably, TaylorSeer can only process Wan 2.1 with 9 frames at 480P and HunyuanVideo with 17 frames at 544P, which are the maximum video lengths that TaylorSeer can process on a single NVIDIA A800 80GB GPU.
>
>
>    | Model | Parameter | Original | PAB | TeaCache | FasterCache | ProfilingDiT | TaylorSeer | BWCache |
>    |-------|-----------|----------|-----|----------|-------------|--------------|------------|---------|
>    | Open-Sora | 51 frames, 480P | 15,270 MiB | 27,016 MiB | 20,630 MiB | 26,130 MiB | - | - | **21,362 MiB** |
>    | Open-Sora-Plan | 65 frames, 512×512 | 15,724 MiB | 26,722 MiB | 21,266 MiB | 22,508 MiB | - | - | **18,022 MiB** |
>    | Latte | 16 frames, 512×512 | 15,240 MiB | 29,126 MiB | 18,499 MiB | 23,264 MiB | - | - | **17,400 MiB** |
>    | Wan 2.1 | 9 frames, 480P | 49,757 MiB | 59,757 MiB | 56,737 MiB | - | 57,089 MiB | 79,263 MiB | **58,047 MiB** |
>    | Wan 2.1 | 81 frames, 480P | 71,861 MiB | 80,285 MiB | 74,197 MiB | - | 78,787 MiB | OOM | **79,541 MiB** |
>    | Wan 2.1 | 81 frames, 720P | 73,955 MiB | 80,154 MiB | 76,563 MiB | - | OOM | OOM | **80,607 MiB** |
>    | HunyuanVideo | 17 frames, 720P | 45,124 MiB | 57,411 MiB | 52,451 MiB | - | 65,124 MiB | 74,287 MiB | **49,757 MiB** |
>    | HunyuanVideo | 129 frames, 540P | 58,491 MiB | 78,787 MiB | 64,729 MiB | - | OOM | OOM | **66,184 MiB** |
>    | HunyuanVideo | 129 frames, 720P | 74,479 MiB | OOM | 78,039 MiB | - | OOM | OOM | **80,625 MiB** |
>
> The results demonstrate that BWCache achieves competitive memory efficiency compared to other cache-based methods.

---

> ### Author Response · Authors · 2025-11-27
> **Looking forward to the reply**
>
> Dear reviewer wT4r:
>
> Thanks so much again for the time and effort in our work. According to the comments and concerns, we conduct the corresponding experiments and further discuss the related points. Additionally, we have added all improvements including 6 detailed experiments in the latest pdf revision.
>
> As the rebuttal period is about to close, may I know if our rebuttal addresses the concerns? If there are further concerns or questions, we are welcome to address them. Thanks again for taking the time to review our work and provide insightful comments.

---

### Official Review · Reviewer_RYjh · 2025-10-31

**Soundness:** 3
**Presentation:** 3
**Contribution:** 3
**Rating:** 6
**Confidence:** 3

**Summary:**

This paper proposes BWCache, a training-free acceleration framework for Diffusion Transformer (DiT)–based video diffusion models. Instead of modifying the model architecture, BWCache reuses intermediate block-level features across diffusion timesteps when the change between adjacent blocks is below a similarity threshold δ. A periodic recomputation strategy mitigates latent drift.

Comprehensive experiments across five representative video diffusion models show that BWCache achieves up to 2.6× speed-up while maintaining comparable or slightly better perceptual quality compared to TeaCache and PAB. Statistical analyses and ablations (threshold $\delta$, reuse interval, GPU scaling) reinforce the empirical soundness.

While the acceleration results are impressive, the approach incurs non-negligible memory overhead due to feature caching, and its stability across architectures and motion intensities remains insufficiently explored. Nevertheless, the work offers a practical plug-and-play contribution that meaningfully advances inference-time efficiency for large-scale DiT-based video generation.

**Strengths:**

BWCache demonstrates outstanding performance in **redundancy granularity**, **cross-scenario adaptability**, and **quality–efficiency balance**, providing a precise and general acceleration framework tailored for redundant computation in DiT-based video generation models.

1. **Focusing on Module-Level Redundancy, Breaking the Granularity Limitation of Traditional Caching**
   Existing caching methods (e.g., TeaCache, PAB) typically operate at the *timestep level* or *attention-head level*: the former risks losing critical timestep information, while the latter incurs high computational overhead and limited acceleration:
   - **Key discovery of module-level redundancy:** By analyzing DiT model computation time distribution, the authors find that DiT modules account for over 70% of total inference latency, and that module features exhibit a *U-shaped variation pattern* across diffusion timesteps — large differences in early (high-noise) and late (low-noise) stages (requiring computation), but small differences in mid-stages (moderate noise, high redundancy, suitable for caching).
   - **Relative L1 distance as a quantitative criterion:** They define the *relative L1 distance* between module features and aggregate it into
     $ \text{ARL1}(t) $ — the sum of all module-wise relative L1 values — as the redundancy measure. When
     $ \text{ARL1}(t)/N < \delta $ (δ = threshold), cached features are reused, ensuring that computation is skipped *only* when true redundancy exists, preventing blind reuse and quality degradation.
   This design focuses on the *most time-consuming components* (modules), guaranteeing both substantial acceleration and controlled quality degradation — the key to BWCache’s superior performance.

2. **Consistent Cross-Model Acceleration with Negligible Quality Loss**
   Across five mainstream video generation models (Open-Sora, Open-Sora-Plan, Latte, Wan 2.1, and HunyuanVideo), BWCache achieves a strong balance between *speed* and *quality*, without requiring retraining:
   - **Single-GPU scenarios:**
     - On *HunyuanVideo* (129 frames, 544 P), BWCache achieves 2.6× speed-up (latency 1122 s → 433 s), with VBench 82.48% (higher than baseline 82.29%), LPIPS 0.0794 (better than PAB 0.1045 and TeaCache 0.1630), and PSNR 29.91.
     - On *Open-Sora*, at 1.61× acceleration, LPIPS 0.0879 (close to the original), SSIM 0.8854, and PSNR 27.05 — all significantly outperforming baselines like Δ-DiT and T-GATE.
   - **Multi-GPU scaling:**
     With *Dynamic Sequence Parallelism (DSP)*, BWCache reaches 17.2× acceleration on 8 A800 GPUs for Open-Sora (204 frames, 480 P) (latency 11.1 s vs 190.2 s original), with near-linear scaling, enabling large-scale deployment.
   - **High-resolution / long-video adaptability:**
     On *Wan 2.1* (720 P), BWCache delivers 2× speed-up (latency 912 s → 457 s) with VBench 81.99% (only 0.18% lower than original), while TeaCache at 1.41× acceleration shows more noticeable quality loss.

**Weaknesses:**

Despite its impressive performance, BWCache still has room for improvement in **memory overhead** and **adaptability to dynamic scenes**.

1. **Module Caching Increases Memory Consumption — Limited for Single-GPU High-Resolution Scenarios**
   BWCache caches the *output features of each DiT module*, leading to noticeable GPU memory growth, especially for large models and high-resolution videos:
   - Appendix D.4 reports that on Wan 2.1 (81 frames, 720 P), BWCache uses 80 607 MiB vs 73 955 MiB for the baseline (+9%), and on HunyuanVideo (129 frames, 720 P) +8%.
   - To reduce memory pressure, the authors compute caching metrics on “50% module sampling”, which lowers memory cost but slightly reduces redundancy detection accuracy (e.g., 10% sampling yields LPIPS 0.7901 vs 0.0782 for 50% sampling).
   This limits BWCache’s applicability for single-GPU, memory-constrained environments (e.g., edge devices or high-resolution video generation).

2. **Insufficient Redundancy Detection Accuracy in Highly Dynamic Scenes**
   The “U-shaped feature variation” assumption holds well for static scenes (e.g., “beach sunset”) but weakens in highly dynamic cases (e.g., “car chase,” “crowded street”):
   - In dynamic scenes (Prompt 2: “bus approaching traffic light”), module-level differences remain larger, with mid-stage redundant steps 30–40% fewer than in static scenes, leading to lower acceleration (e.g., on Open-Sora-Plan, TeaCache achieves 4.41× while BWCache only 2.24×, Table 1).
   - No dynamic-aware adaptation is proposed: the method lacks an *adaptive threshold $\delta$* based on scene dynamics (e.g., using smaller δ for fast-changing content), resulting in suboptimal speed-quality trade-offs in dynamic video generation.

**Questions:**

1. Are spatial and temporal DiT blocks cached jointly or independently, and did you observe temporal drift in high-motion clips
2. Can you provide results on highly dynamic or compositional scenes (human motion, fast parallax) to test stability

---

> ### Author Response · Authors · 2025-11-23
> **Response to Reviewer RYjh (1/2)**
>
> **_W1: Module Caching Increases Memory Consumption — Limited for Single-GPU High-Resolution Scenarios_**
>
> **Response:** Thanks for the comment. All training-free caching methods require additional GPU memory compared to the original models due to caching intermediate features from previous timesteps.
>
> The detailed memory usage results are shown in Table 12. Notably, TaylorSeer can only process Wan 2.1 with 9 frames at 480P and HunyuanVideo with 17 frames at 544P, which are the maximum video lengths that TaylorSeer can process on a single NVIDIA A800 80GB GPU.
>
>    | Model | Parameter | Original | PAB | TeaCache | FasterCache | ProfilingDiT | TaylorSeer | BWCache |
>    |-------|-----------|----------|-----|----------|-------------|--------------|------------|---------|
>    | Open-Sora | 51 frames, 480P | 15,270 MiB | 27,016 MiB | 20,630 MiB | 26,130 MiB | - | - | **21,362 MiB** |
>    | Open-Sora-Plan | 65 frames, 512×512 | 15,724 MiB | 26,722 MiB | 21,266 MiB | 22,508 MiB | - | - | **18,022 MiB** |
>    | Latte | 16 frames, 512×512 | 15,240 MiB | 29,126 MiB | 18,499 MiB | 23,264 MiB | - | - | **17,400 MiB** |
>    | Wan 2.1 | 9 frames, 480P | 49,757 MiB | 59,757 MiB | 56,737 MiB | - | 57,089 MiB | 79,263 MiB | **58,047 MiB** |
>    | Wan 2.1 | 81 frames, 480P | 71,861 MiB | 80,285 MiB | 74,197 MiB | - | 78,787 MiB | OOM | **79,541 MiB** |
>    | Wan 2.1 | 81 frames, 720P | 73,955 MiB | 80,154 MiB | 76,563 MiB | - | OOM | OOM | **80,607 MiB** |
>    | HunyuanVideo | 17 frames, 720P | 45,124 MiB | 57,411 MiB | 52,451 MiB | - | 65,124 MiB | 74,287 MiB | **49,757 MiB** |
>    | HunyuanVideo | 129 frames, 540P | 58,491 MiB | 78,787 MiB | 64,729 MiB | - | OOM | OOM | **66,184 MiB** |
>    | HunyuanVideo | 129 frames, 720P | 74,479 MiB | OOM | 78,039 MiB | - | OOM | OOM | **80,625 MiB** |
>
>
> Importantly, BWCache is not an all-or-nothing design; its speed-memory trade-off is configurable and can be adapted for memory-constrained settings. For example, only cache a subset of blocks instead of all blocks. The detailed results for caching a subset of blocks are shown in Table 13.
>
> **Wan 2.1 (81 frames, 480P)**
>
> | Blocks Used | Memory Usage | LPIPS↓ | SSIM↑ | PSNR↑ |
> |-------------|--------------|--------|-------|-------|
> | 10% blocks | 72367 MiB | 0.0790 | 0.8524 | 25.79 |
> | 30% blocks | 75621 MiB | 0.0789 | 0.8533 | 25.83 |
> | 50% blocks | 79541 MiB | 0.0782 | 0.8539 | 25.86 |
>
> **HunyuanVideo (129 frames, 540P)**
>
> | Blocks Used | Memory Usage | LPIPS↓ | SSIM↑ | PSNR↑ |
> |-------------|--------------|--------|-------|-------|
> | 10% blocks | 60977 MiB | 0.0795 | 0.8921 | 29.95 |
> | 30% blocks | 63617 MiB | 0.0786 | 0.8884 | 29.86 |
> | 50% blocks | 66184 MiB | 0.0794 | 0.8903 | 29.91 |
>
>
> These results demonstrate that BWCache can effectively balance memory efficiency and generation quality by adjusting the number of cached blocks, making it adaptable to various hardware constraints while maintaining competitive performance.

---

> ### Author Response · Authors · 2025-11-23
> **Response to Reviewer RYjh (2/2)**
>
> **_W2: Insufficient Redundancy Detection Accuracy in Highly Dynamic Scenes._**
>
>
> **Response:** Thank you for raising this important concern about dynamic scenes. However, we believe there may be some misunderstanding regarding our method's behavior in dynamic scenarios.
>
> **First**, the U-shaped pattern of feature variations is a universal characteristic of the diffusion process across all types of scenes. Therefore, the key distinction is not whether the U-shaped pattern exists, but rather the duration of the stable region at the bottom of the U-shape. In dynamic scenes, the stable region is shorter, but the pattern itself remains valid.
>
> **Second**, we acknowledge that TeaCache achieves 4.41× speedup on Open-Sora-Plan compared to BWCache's 2.24×. However, this comparison overlooks the crucial quality aspect. TeaCache's higher speedup comes at a significant quality cost: it has higher LPIPS, lower SSIM, and lower PSNR.
>
> **Last**, regarding the concern about lacking dynamic-aware adaptation, we would like to clarify that BWCache already implements dynamic, block-level adaptive caching. As we explain in Section 3.3:
>
> > "The threshold $\delta$ represents an acceptable level of variation, and when features change more, fewer blocks are cached, naturally adapting to the scene dynamics."
>
>
>
> **_Q1: Are spatial and temporal DiT blocks cached jointly or independently, and did you observe temporal drift in high-motion clips._**
>
> **Response:** Thanks for the comment. In BWCache, spatial and temporal DiT blocks are cached independently. This independent caching strategy enables our method to adapt more effectively to the different dynamics exhibited by spatial and temporal features during the denoising process. In addition, BWCache effectively mitigates temporal drift through adaptive caching, periodic recomputation, and careful final-step refinement, even in highly dynamic scenarios.
>
>
> **_Q2: Can you provide results on highly dynamic or compositional scenes (human motion, fast parallax) to test stability._**
>
> **Response:** Thank you for your suggestion. We would like to clarify that our comprehensive experiments have already evaluated BWCache across a wide range of scenarios, including highly dynamic and compositional scenes. To directly address your concern and provide a focused analysis, we have specifically extracted and evaluated a subset of videos that exhibit high dynamic characteristics from our comprehensive evaluation. As shown in Table 14 and Figure 11, we have evaluated BWCache's performance on highly dynamic video generation tasks across five models.
>
>
>    | Model | Original VBench | BWCache VBench |
>    |-------|-----------------|----------------|
>    | Open-Sora | 80.12% | 79.71% |
>    | Open-Sora-Plan | 80.48% | 80.37% |
>    | Latte | 78.23% | 77.64% |
>    | Wan 2.1 | 82.03% | 81.90% |
>    | HunyuanVideo | 82.14% | 81.38% |

---

> ### Author Response · Authors · 2025-11-27
> **Looking forward to the reply**
>
> Dear reviewer RYjh:
>
> Thanks so much again for the time and effort in our work. According to the comments and concerns, we conduct the corresponding experiments and further discuss the related points. Additionally, we have added all improvements including 6 detailed experiments in the latest pdf revision.
>
> As the rebuttal period is about to close, may I know if our rebuttal addresses the concerns? If there are further concerns or questions, we are welcome to address them. Thanks again for taking the time to review our work and provide insightful comments.

---

### Official Review · Reviewer_41nC · 2025-11-01

**Soundness:** 3
**Presentation:** 3
**Contribution:** 3
**Rating:** 6
**Confidence:** 4

**Summary:**

This paper presents BWCache, a training-free inference acceleration method for DiT-based video generation. The work is motivated by an empirical analysis showing that DiT block computations are the main latency bottleneck and exhibit a U-shaped variation pattern, implying high redundancy in intermediate steps. BWCache exploits this by dynamically caching and reusing features at the block level. A similarity indicator (average relative L1 distance) determines when to reuse features, while safeguards like periodic recomputation and forced computation in the final steps ensure high visual fidelity. The method is validated across five state-of-the-art models, demonstrating significant speedups while consistently outperforming other acceleration baselines in preserving video quality.

**Strengths:**

1. The paper is built on a clear, insightful, and well-visualized analysis (Figure 4). The identification of the U-shaped variation pattern in DiT blocks is a strong finding that provides a direct and logical justification for the entire method's design.
2. The method is more sophisticated than several competitors. It correctly identifies a suitable *granularity* (DiT blocks) and, more importantly, implements a dynamic *trigger* (the similarity indicator) for caching. This, combined with the fidelity-preserving strategies (final-step recomputation), allows it to effectively balance acceleration and quality.
3. The experimental validation is thorough. Testing on five different, large-scale video DiTs and comparing against multiple relevant baselines demonstrates the method's general applicability. The scalability tests (multi-GPU, video length) and the direct ablation against naive step reduction (Table 3) make the performance claims very convincing.

**Weaknesses:**

1. Notably, the trade-off of speed for memory is a weakness. By caching DiT block features, the method's GPU memory footprint increases, as acknowledged in the appendix (Appendix D.4). For very large models that already push the limits of high-end GPUs (e.g., Wan 2.1 using ~74GB), this additional overhead (to ~81GB) could make the method unusable on memory-constrained hardware.
2. The method's effectiveness depends on manually tuning two key hyperparameters: the similarity threshold $\delta$ and the recomputation interval $R$. The ablation studies show that these values control a sensitive trade-off between speed and quality. While the authors provide default values, these may not be optimal for all models or use cases, and the paper does not offer a strategy for setting them automatically.
3. A critical implementation detail is hidden in the appendix (D.4). For memory-heavy models (Wan 2.1, HunyuanVideo), the similarity indicator is not calculated using *all* blocks; instead, a *subset* (e.g., 50%) is used to save memory. The main paper does not mention this, and the appendix does not specify *how* this subset is chosen (e.g., first 50%, random, etc.), making the indicator's true behavior and robustness on these models unclear.
4. The fidelity safeguards are static: always recompute the *last $k/2$ steps* and reuse for a *fixed interval $R$*. The "U-shape" analysis suggests that the transition from stable to variable features (the right side of the "U") is gradual. A more dynamic approach, perhaps by monitoring the L1 distance to decide when to *stop* caching (rather than a fixed $k/2$ rule), might offer an even better trade-off.

**Questions:**

What is the underlying theoretical explanation for the observed U-shaped pattern in DiT block feature variations? Is this pattern consistent across different architectures and training methodologies, and how does it relate to the diffusion process dynamics? I would appreciate it if the authors could discuss this with previous related works (e.g., [1])

[1] Ma, Xuran, et al. "Model Reveals What to Cache: Profiling-Based Feature Reuse for Video Diffusion Models." Proceedings of the IEEE/CVF International Conference on Computer Vision (2025).

---

> ### Author Response · Authors · 2025-11-23
> **Response to Reviewer 41nC (1/3)**
>
> **_W1: Notably, the trade-off of speed for memory is a weakness. For very large models that already push the limits of high-end GPUs, this additional overhead could make the method unusable on memory-constrained hardware._**
>
> **Response:** Thanks for the comment. All training-free caching methods require additional GPU memory compared to the original models due to caching intermediate features from previous timesteps. The detailed memory usage results are shown in Table 12.
>
>    | Model | Parameter | Original | PAB | TeaCache | FasterCache | ProfilingDiT | TaylorSeer | BWCache |
>    |-------|-----------|----------|-----|----------|-------------|--------------|------------|---------|
>    | Open-Sora | 51 frames, 480P | 15,270 MiB | 27,016 MiB | 20,630 MiB | 26,130 MiB | - | - | **21,362 MiB** |
>    | Open-Sora-Plan | 65 frames, 512×512 | 15,724 MiB | 26,722 MiB | 21,266 MiB | 22,508 MiB | - | - | **18,022 MiB** |
>    | Latte | 16 frames, 512×512 | 15,240 MiB | 29,126 MiB | 18,499 MiB | 23,264 MiB | - | - | **17,400 MiB** |
>    | Wan 2.1 | 9 frames, 480P | 49,757 MiB | 59,757 MiB | 56,737 MiB | - | 57,089 MiB | 79,263 MiB | **58,047 MiB** |
>    | Wan 2.1 | 81 frames, 480P | 71,861 MiB | 80,285 MiB | 74,197 MiB | - | 78,787 MiB | OOM | **79,541 MiB** |
>    | Wan 2.1 | 81 frames, 720P | 73,955 MiB | 80,154 MiB | 76,563 MiB | - | OOM | OOM | **80,607 MiB** |
>    | HunyuanVideo | 17 frames, 720P | 45,124 MiB | 57,411 MiB | 52,451 MiB | - | 65,124 MiB | 74,287 MiB | **49,757 MiB** |
>    | HunyuanVideo | 129 frames, 540P | 58,491 MiB | 78,787 MiB | 64,729 MiB | - | OOM | OOM | **66,184 MiB** |
>    | HunyuanVideo | 129 frames, 720P | 74,479 MiB | OOM | 78,039 MiB | - | OOM | OOM | **80,625 MiB** |
>
>
> Importantly, BWCache is not an all-or-nothing design; its speed-memory trade-off is configurable and can be adapted for memory-constrained settings. For example, only cache a subset of blocks instead of all blocks. The detailed results for caching a subset of blocks are shown in Table 13.
>
> **Wan 2.1 (81 frames, 480P)**
>
> | Blocks Used | Memory Usage | LPIPS↓ | SSIM↑ | PSNR↑ |
> |-------------|--------------|--------|-------|-------|
> | 10% blocks | 72367 MiB | 0.0790 | 0.8524 | 25.79 |
> | 30% blocks | 75621 MiB | 0.0789 | 0.8533 | 25.83 |
> | 50% blocks | 79541 MiB | 0.0782 | 0.8539 | 25.86 |
>
> **HunyuanVideo (129 frames, 540P)**
>
> | Blocks Used | Memory Usage | LPIPS↓ | SSIM↑ | PSNR↑ |
> |-------------|--------------|--------|-------|-------|
> | 10% blocks | 60977 MiB | 0.0795 | 0.8921 | 29.95 |
> | 30% blocks | 63617 MiB | 0.0786 | 0.8884 | 29.86 |
> | 50% blocks | 66184 MiB | 0.0794 | 0.8903 | 29.91 |
>
>
> These results demonstrate that BWCache can effectively balance memory efficiency and generation quality by adjusting the number of cached blocks, making it adaptable to various hardware constraints while maintaining competitive performance.

---

> ### Author Response · Authors · 2025-11-23
> **Response to Reviewer 41nC (2/3)**
>
> **_W2: While the authors provide default values, these may not be optimal for all models or use cases, and the paper does not offer a strategy for setting them automatically._**
>
> **Response:** Thanks for the comment. While not fully automatic, our method has proven that $\delta = 0.15$ and $R = 10\%$ perform well on most models. Additionally, Table 15 provides model-specific threshold ranges that have been validated across five different models:
>
>    | Model | Threshold Range | VBench Range | Speedup Range |
>    |-------|----------------|--------------|---------------|
>    | Open-Sora | 0.15 - 0.25 | 79.3\% - 80.0\% | 1.41 - 1.96× |
>    | Open-Sora-Plan | 0.15 - 0.25 | 80.2\% - 81.0\% | 2.02 - 2.52× |
>    | Latte | 0.1 - 0.2 | 75.5\% - 78.7\% | 1.67 - 2.07× |
>    | Wan 2.1 | 0.1 - 0.2 | 80.4\% - 81.3\% | 1.75 - 2.48× |
>    | HunyuanVideo | 0.15 - 0.25 | 81.9\% - 82.6\% | 2.31 - 2.94× |
>
>
>
> We have added guidelines for the parameter in the revised manuscript (Appendix E.1; Page 21).
>
>    > "When applying BWCache to a new DiT-based model, we recommend starting with a default value of $\delta = 0.15$, which prioritizes quality. To further optimize, generate a few sample videos and plot the ARL1 across timesteps to identify the optimal region for cache reuse, then iteratively test different threshold values to balance quality and speedup. Models with more stable feature dynamics may tolerate higher thresholds, while those handling highly dynamic content may require lower thresholds. During threshold selection, prioritize maintaining VBench scores within 1-2\% of the original model's performance, and reduce the threshold if quality degradation exceeds acceptable limits."
>
>
>
>
> **_W3: The main paper does not mention this, and the appendix does not specify how this subset is chosen (e.g., first 50%, random, etc.)._**
>
> **Response:** Thank you for this important observation. We have now explicitly clarified the subset selection method in the revised manuscript (Appendix D.5; Page 18).
>
>    > "We compute the similarity indicator using a uniformly spaced subset of blocks across depth. This design aims to cover both early, middle, and late blocks while reducing the amount of cached state and indicator computation."
>
> **_W4: A more dynamic approach, perhaps by monitoring the L1 distance to decide when to stop caching (rather than a fixed rule), might offer an even better trade-off._**
>
> **Response:** Thanks for the comment. We have indeed explored a more dynamic approach that monitors the L1 distance to decide when to stop caching. However, our analysis reveals that once cache reuse begins, feature changes become unpredictable and exhibit irregular patterns, making it very difficult to design or benefit from a more dynamic stopping strategy. We have updated this discussion in the revised manuscript (Appendix E.3; Page 21).
>
> Consequently, we choose a static approach with fixed recomputation intervals that provides more stable and reliable performance.

---

> ### Author Response · Authors · 2025-11-23
> **Response to Reviewer 41nC (3/3)**
>
> **_Q1-1: What is the underlying theoretical explanation for the observed U-shaped pattern in DiT block feature variations?_**
>
> **Response:** Thanks for the comment. The theoretical foundation for the U-shaped pattern can be understood through frequency domain analysis of the diffusion process [1]. During early timesteps, low-frequency components recover, causing high feature variation (left U-shape). Middle timesteps stabilize these, minimizing changes (U-bottom). Final timesteps recover high-frequency details, increasing variation again (right U-shape).
>
>
> [1] Yurui Qian, et al. "Boosting Diffusion Models with Moving Average Sampling in Frequency Domain." In Proceedings of the IEEE/CVF Conference on Computer Vision and Pattern Recognition (CVPR), 2024.
>
> **_Q1-2: Is this pattern consistent across different architectures and training methodologies, and how does it relate to the diffusion process dynamics?_**
>
> **Response:** Thanks for the comment. As discussed in our response to Q1-1, the frequency-domain perspective explains why this pattern emerges regardless of the specific DiT architecture or training approach. However, we do observe some variations in the intensity and timing of the pattern. Specifically, Open-Sora-Plan uses non-linear samplers (i.e. PNDM) that do not start from the last timestep ($t=T$), creating a training-inference discrepancy where the model is forced to handle inputs that deviate from pure Gaussian noise, potentially causing initial timestep feature prediction instability and oscillations [1].
>
>
> [1] Shanchuan Lin, et al. "Common Diffusion Noise Schedules and Sample Steps are Flawed." arXiv preprint arXiv:2305.08891, 2024.
>
> **_Q1-3: I would appreciate it if the authors could discuss this with previous related works (e.g., [1]) \\  [1] Ma, Xuran, et al. "Model Reveals What to Cache: Profiling-Based Feature Reuse for Video Diffusion Models." Proceedings of the IEEE/CVF International Conference on Computer Vision (2025)._**
>
> **Response:** Thanks for the comment. ProfilingDiT indicates that redundancy is not uniformly distributed across timesteps. While their work does not focus on block-wise analysis in DiTs, their findings are conceptually aligned with our observation that mid-timestep regions exhibit higher redundancy.
>
> Furthermore, we have conducted experimental comparisons with ProfilingDiT. Notably, ProfilingDiT encounters out-of-memory (OOM) errors when generating long videos. To ensure a fair comparison, we evaluate these methods on shorter video sequences. The parameter $T_s$ for ProfilingDiT is set to 6 both for Wan 2.1 and HunyuanVideo. The detailed experimental results are shown in Table 7:
>
>    | Model | Method | VBench↑ | LPIPS↓ | SSIM↑ | PSNR↑ | Speedup↑ | Latency(s)↓ |
>    |-------|--------|---------|--------|-------|-------|----------|-------------|
>    | Wan 2.1 (9 frames, 480P) | ProfilingDiT | 76.53% | 0.2053 | 0.8221 | 24.20 | 1.24× | 231 |
>    | | BWCache | **76.81%** | **0.0929** | **0.9048** | **29.65** | **1.67×** | **171** |
>    | HunyuanVideo (17 frames, 544P) | ProfilingDiT | 80.33% | 0.2610 | 0.6662 | 18.77 | 1.72× | 104 |
>    | | BWCache | **81.42%** | **0.1804** | **0.7507** | **21.60** | **2.39×** | **75** |

---

> ### Author Response · Authors · 2025-11-27
> **Looking forward to the reply**
>
> Dear reviewer 41nC:
>
> Thanks so much again for the time and effort in our work. According to the comments and concerns, we conduct the corresponding experiments and further discuss the related points. Additionally, we have added all improvements including 6 detailed experiments in the latest pdf revision.
>
> As the rebuttal period is about to close, may I know if our rebuttal addresses the concerns? If there are further concerns or questions, we are welcome to address them. Thanks again for taking the time to review our work and provide insightful comments.

---

### Official Review · Reviewer_CKpW · 2025-11-04

**Soundness:** 3
**Presentation:** 3
**Contribution:** 3
**Rating:** 4
**Confidence:** 4

**Summary:**

The paper proposes BWCache, a training-free acceleration technique for Diffusion Transformer video generation. It starts with the observation that DiT blocks dominate latency and many intermediate timesteps present redundant computation. BWCache caches block outputs across timesteps and reuses them based on a similarity indicator. It also incorporates periodic recomputation to prevent latent drift. Experiments across multiple backbone models show up to 2.6× speedup while maintaining competitive quality.

**Strengths:**

- High Practicality and Scalability. As a training-free method, it can be integrated into existing DiT pipelines without costly fine-tuning.
- The BWCache design is well-motivated by a clear empirical analysis of DiT block behavior. The Aggregated Relative L1 analysis in Figure 4 provides a strong justification for the caching strategy.
- The method achieves a superior balance between inference speed and visual quality. It delivers speedups while maintaining comparable or even superior visual fidelity.

**Weaknesses:**

- The paper's claim to be the first method to exploit block-wise temporal stability in DiTs appears to be overstated. There is some highly relevant prior work which also investigates temporal redundancy in DiTs but not discussed in the paper. The authors must situate their work relative to it, differentiate their contribution, and (if possible) provide a comparative analysis.
1. Towards Stabilized and Efficient Diffusion Transformers through Long-Skip-Connections with Spectral Constraints

- In Table 1, TeaCache achieves a 4.41x speedup on Open-Sora-Plan. The authors attribute it to "significant variation in block features during the early inference stages," but provide no data or analysis to support this claim. Why does the block-wise indicator in Eq. 7 fail in this specific scenario while TeaCache's simpler timestep-level caching succeeds?
-  The entire caching mechanism depends on a manually-set similarity threshold in Eq. 7, which appears empirical. The paper provides no theoretical grounding for this choice, nor does it analyze its stability or the risk of error accumulation from repeated cache reuse. The ablation in Table 4 shows a predictable quality/speed trade-off but fails to explain how a user should set $\delta$ for a new model or task.
- The computational cost of the cache indicator is not considered. BWCache requires calculating the L1 distance for all blocks at every step to decide whether to cache. This overhead is never measured. While this cost may be negligible on large models, it could substantially diminish on smaller models, in edge-computing scenarios, or on hardware with lower memory bandwidth.

**Questions:**

The paper's validation is limited exclusively to text-to-video generation. Have you explore the core "U-shaped" ARL1 pattern holds in other settings, like class-conditional generation, image generation?

---

> ### Author Response · Authors · 2025-11-23
> **Response to Reviewer CKpW (1/3)**
>
> **_W1: The paper\'s claim to be the first method to exploit block-wise temporal stability in DiTs appears to be overstated. There is some highly relevant prior work which also investigates temporal redundancy in DiTs but not discussed in the paper. The authors must situate their work relative to it, differentiate their contribution, and (if possible) provide a comparative analysis : Towards Stabilized and Efficient Diffusion Transformers through Long-Skip-Connections with Spectral Constraints_**
>
> **Response:** Thanks for the comment. We would like to clarify that our paper does not claim to be the first method to exploit block-wise temporal stability in DiTs. In addition, Skip-DiT is not a training-free acceleration method, and thus was not included as a baseline before. According to your suggestion, we have updated this related work in the revised manuscript (Section 2.2; Page 3).
>
>
>    > "Skip-DiT addresses feature instability by introducing Long-Skip-Connections to stabilize DiT block features, enabling efficient caching. However, this method requires modifying the model architecture and retraining from scratch, which is impractical for large pre-trained models and limits their applicability.
>
>    > "Our method is training-free and memory-efficient, using a lightweight similarity indicator to dynamically determine cache reuse without extensive feature storage, making it directly applicable to existing pre-trained models without architectural modifications."
>
>
> **Comparative Analysis:** While Skip-DiT achieves competitive acceleration, it requires architectural modifications and model retraining, which significantly limit its practical applicability. The Skip-DiT repository only provides a pre-trained model for Latte, and according to their documentation, training requires approximately 8 H100 GPUs for about one week with 300k text-video pairs. This substantial training requirement makes it impractical to evaluate Skip-DiT on other models or compare it fairly across different architectures. We have added Skip-DiT to Table 1 for comparison on the Latte model.
>
>
>    | Model | Method | VBench↑ | LPIPS↓ | SSIM↑ | PSNR↑ | Speedup↑ | Latency(s)↓ |
>    |--------|--------|---------|--------|-------|-------|----------|-------------|
>    | Latte (16 frames, 512×512) | Skip-DiT | 75.76% | 0.1403 | 0.8087 | 25.50 | 1.65× | 16.28 |
>    | | BWCache | **78.28%** | **0.1399** | **0.8181** | **26.46** | **1.90×** | **14.16** |
>
>
> **_W2: Why does the block-wise indicator in Eq. 7 fail in this specific scenario while TeaCache's simpler timestep-level caching succeeds?_**
>
> **Response:** Thanks for the comment. Open-Sora-Plan exhibits more pronounced early-stage feature oscillations compared to other models. We have updated this discussion in the revised manuscript (Section 3.2; Page 5).
>
> > "This can be attributed to noise schedule and sampling implementation issues [1]. Specifically, Open-Sora-Plan uses non-linear samplers (i.e. PNDM) that do not start from the last timestep ($t=T$), creating a training-inference discrepancy where the model is forced to handle inputs that deviate from pure Gaussian noise, potentially causing initial timestep feature prediction instability and oscillations."
>
> To maintain high-fidelity output quality, BWCache strategically skips caching during the first few denoising steps where feature oscillations are most pronounced, thereby sacrificing some acceleration potential in favor of generation quality.
>
> [1] Shanchuan Lin, et al. "Common Diffusion Noise Schedules and Sample Steps are Flawed." arXiv preprint arXiv:2305.08891, 2024.

---

> ### Author Response · Authors · 2025-11-23
> **Response to Reviewer CKpW (2/3)**
>
> **_W3-1: The entire caching mechanism depends on a manually-set similarity threshold in Eq. 7, which appears empirical._**
>
> **Response:** Thank you for these valuable comments. The choice of the relative L1 distance as the similarity metric and the threshold $\delta$ is grounded in the U-shaped pattern of feature variations across diffusion timesteps. This U-shaped pattern is consistent across different architectures and training methodologies, as validated in Figure 5. The threshold $\delta$ captures the stability region during intermediate timesteps where feature variations are minimal, allowing safe cache reuse.
>
> **_W3-2: The paper provides no theoretical grounding for this choice, nor does it analyze its stability or the risk of error accumulation from repeated cache reuse._**
>
> **Response:** Thank you for these valuable comments. To solve the problem of error accumulation from repeated cache reuse, we adopt the periodic recomputation strategy to prevent error accumulation. Additionally, our design always computes the last $k/2$ steps, ensuring that any accumulated errors are corrected during the final refinement phase.
>
> **_W3-3: The ablation in Table 4 shows a predictable quality/speed trade-off but fails to explain how a user should set $\delta$ for a new model or task._**
>
> **Response:** Thank you for pointing out this important practical concern. Usually, our method has proven that $\delta = 0.15$ and $R = 10\%$ perform well on most models. We have added comprehensive guidelines for setting the threshold $\delta$ for new models in the revised manuscript (Appendix E.1; Page 20).
>
>
>    > "When applying BWCache to a new DiT-based model, we recommend starting with a default value of $\delta = 0.15$, which prioritizes quality. To further optimize, generate a few sample videos and plot the ARL1 across timesteps to identify the optimal region for cache reuse, then iteratively test different threshold values to balance quality and speedup. Models with more stable feature dynamics may tolerate higher thresholds, while those handling highly dynamic content may require lower thresholds. During threshold selection, prioritize maintaining VBench scores within 1-2\% of the original model's performance, and reduce the threshold if quality degradation exceeds acceptable limits."
>
>
>
> Additionally, Table 15 in the appendix provides model-specific threshold ranges that have been validated across five different models:
>
>    | Model | Threshold Range | VBench Range | Speedup Range |
>    |-------|----------------|--------------|---------------|
>    | Open-Sora | 0.15 - 0.25 | 79.3\% - 80.0\% | 1.41 - 1.96× |
>    | Open-Sora-Plan | 0.15 - 0.25 | 80.2\% - 81.0\% | 2.02 - 2.52× |
>    | Latte | 0.1 - 0.2 | 75.5\% - 78.7\% | 1.67 - 2.07× |
>    | Wan 2.1 | 0.1 - 0.2 | 80.4\% - 81.3\% | 1.75 - 2.48× |
>    | HunyuanVideo | 0.15 - 0.25 | 81.9\% - 82.6\% | 2.31 - 2.94× |

---

> ### Author Response · Authors · 2025-11-23
> **Response to Reviewer CKpW (3/3)**
>
> **_W4: The computational cost of the cache indicator is not considered._**
>
> **Response:** Thanks for the comment. The computational cost of the cache indicator encompasses two main aspects: memory usage and computational latency.
>
> **Memory Usage:** The detailed memory usage results are shown in Table 11. The additional memory overhead compared to the original model represents the memory cost required to run the algorithm.
>
> **Open-Sora, Open-Sora-Plan, Latte**
>
> | Model | Open-Sora | Open-Sora-Plan | Latte |
> |-------|-----------|----------------|-------|
> | Parameter | 51 frames, 480P | 65 frames, 512 × 512 | 16 frames, 512 × 512 |
> | **Original** | 15270MiB | 15724MiB | 15240MiB |
> | **BWCache** | 21362MiB | 18022MiB | 17400MiB |
>
> **Wan 2.1**
>
> | Model | 9 frames, 480P | 81 frames, 480P | 81 frames, 720P |
> |-------|----------------|-----------------|-----------------|
> | **Original** | 49757MiB | 71861MiB | 73955MiB |
> | **BWCache** | 58047MiB | 79541MiB | 80607MiB |
>
>
> **HunyuanVideo**
>
> | Model | 17 frames, 720P | 129 frames, 540P | 129 frames, 720P |
> |-------|-----------------|------------------|------------------|
> | **Original** | 45124MiB | 58491MiB | 74479MiB |
> | **BWCache** | 49757MiB | 66184MiB | 80625MiB |
>
>
> **Computational Latency:** We have measured the absolute time spent on computing the cache indicator and its relative proportion of the total inference time across five models. The detailed computational latency results are shown in Table 13.
>
>    | Model | Open-Sora | Open-Sora-Plan | Latte | Wan 2.1 | HunyuanVideo |
>    |-------|-----------|----------------|-------|--------|--------------|
>    | **Computational latency (s)** | 6.22(22.5\%) | 5.05(11.4\%) | 3.12(22.0\%) | 48.23(10.6\%) | 68.90(15.9\%) |
>
>
> **_Q1: Have you explore the core "U-shaped" ARL1 pattern holds in other settings, like class-conditional generation, image generation?_**
>
> **Response:** Thank you for your insights. We have indeed explored whether the U-shaped pattern extends to other diffusion model settings. The feature variation pattern is closely related to the noise schedule and loss function used during training [1]. Unfortunately, class-conditional generation and image generation exhibit an inverted L-shaped pattern, which differs from the U-shaped pattern observed in video generation models. We have updated this discussion in the revised manuscript (Appendix E.2; Page 21).
>
> [1] Cross-Attention Makes Inference Cumbersome in Text-to-Image Diffusion Models. arXiv:2404.02747v1.

---

> ### Author Response · Authors · 2025-11-27
> **Looking forward to the reply**
>
> Dear reviewer CKpW:
>
> Thanks so much again for the time and effort in our work. According to the comments and concerns, we conduct the corresponding experiments and further discuss the related points. Additionally, we have added all improvements including 6 detailed experiments in the latest pdf revision.
>
> As the rebuttal period is about to close, may I know if our rebuttal addresses the concerns? If there are further concerns or questions, we are welcome to address them. Thanks again for taking the time to review our work and provide insightful comments.

---

### Author Response · Authors · 2025-11-30
**General Response**

Dear Area Chair,

We would like to express our sincere gratitude for your guidance during the review process. We are also grateful for the constructive feedback from all four reviewers.

**BWCache** is a training-free acceleration method for DiT-based video generation. Our key innovation is identifying a **U-shaped pattern** in DiT feature variations across diffusion timesteps, revealing high redundancy in intermediate timesteps suitable for caching. BWCache achieves **up to 2.6× speedup** across five state-of-the-art models while maintaining or improving visual quality.

All reviewers recognized the practical value of our work. Key strengths highlighted include: (1) **High practicality** as a training-free method requiring no fine-tuning; (2) **Well-motivated design** based on clear empirical and theoretical analysis of the U-shaped pattern; (3) **Superior quality-speed balance** with consistent cross-model acceleration.

Below, we summarize the changes made in the revised paper:

1. Reviewers CKpW and wT4r pointed out the missing comparisons with Skip-DiT, FasterCache, and TaylorSeer. We added Skip-DiT to the related work and clarified that BWCache is training-free, while Skip-DiT requires architectural changes and retraining. We also provided comparative analysis with Skip-DiT showing BWCache's superiority in Table 1. In addition, we added FasterCache and TaylorSeer comparisons in Tables 1 and 7, demonstrating BWCache's better quality-efficiency balance.

2. Reviewers CKpW and 41nC requested explanation of the U-shaped pattern and questioned the theoretical basis for the threshold $\delta$. We provided a frequency-domain theoretical foundation and explained that $\delta$ captures this stability region validated across architectures in Figure 5 and Section 3.2. We further added discussion of ProfilingDiT and experimental comparison in Appendix D.1.

3. Reviewers 41nC, RYjh, and CKpW raised concerns about memory footprint and unmeasured computational cost. We provided comprehensive memory comparison with all baselines in Table 11, demonstrated configurable memory trade-off via subset caching in Table 12, and measured computational cost in Table 13.

4. Reviewers CKpW and 41nC noted lack of guidance for setting hyperparameters on new models. We added comprehensive guidelines with default values, step-by-step instructions, and model-specific threshold ranges validated across all models in Table 15 and Appendix E.1.

5. Reviewer 41nC questioned the subset selection and dynamic cache stopping methods, and Reviewer RYjh requested evaluation on highly dynamic scenes. We clarified the uniformly spaced subset selection strategy in Appendix D.5, explained latent drift from feature reuse in Appendix E.3, and provided dedicated evaluation on highly dynamic scenes showing minimal quality degradation in Appendix D.7.

6. Reviewer CKpW questioned the speedup difference with TeaCache and whether the U-shaped pattern holds in other settings. We explained Open-Sora-Plan's early-stage oscillations due to non-linear samplers in Section 3.2. We explored other settings, explaining that class-conditional and image generation exhibit different patterns in Appendix E.2.

The revised content is highlighted in blue.

We sincerely thank all reviewers again for their valuable suggestions, which have greatly helped strengthen our paper. We are also deeply grateful for your oversight and coordination of the review process.

---

### Meta-Review · Area_Chair_ZwV3 · 2026-01-06

**Summary:**

The reviewers consistently acknowledge the strengths of this work, for example: (1) high practicality and scalability, (2) superior balance between inference speed and visual quality, and (3) the experimental validation is thorough. During the rebuttal, the authors have addressed the major concerns of reviewers by adding missing baseline comparisons (e.g., FasterCache, TaylorSeer, Skip-DiT), providing a frequency-domain explanation for the U-shaped feature variation, and including additional experiments. Based on the reviewers’ comments and the authors’ responses, my recommendation for this submission is acceptance.

**Reviewer Concerns:**

After reviewming the responses, I find that the authors have addressed the major concerns from reviewers by adding missing baseline comparisons (e.g., FasterCache, TaylorSeer, Skip-DiT), providing a frequency-domain explanation for the U-shaped feature variation, and including extensive memory and latency evaluations.

**Reviewer Scores:**

The initial reviewer scores are 6/4/6/6, with three reviews being positive. By carefully reviewing the reviewers’ comments and the authors’ responses, I believe that the reviewers would maintain an overall positive attitude toward this work.

---

### Decision · Program_Chairs · 2026-01-26

Accept (Poster)